# Cortical microtubule nucleation can organise the cytoskeleton of *Drosophila* oocytes to define the anteroposterior axis

Philipp Khuc Trong[1,2], Hélène Doerflinger[3], Jörn Dunkel[1,4], Daniel St Johnston[3], Raymond E Goldstein[1]*

[1]Department of Applied Mathematics and Theoretical Physics, University of Cambridge, Cambridge, United Kingdom; [2]Department of Physics, University of Cambridge, Cambridge, United Kingdom; [3]Wellcome Trust/Cancer Research UK Gurdon Institute, Henry Wellcome Building of Cancer and Developmental Biology, University of Cambridge, Cambridge, United Kingdom; [4]Department of Mathematics, Massachusetts Institute of Technology, Cambridge, United States

**Abstract** Many cells contain non-centrosomal arrays of microtubules (MTs), but the assembly, organisation and function of these arrays are poorly understood. We present the first theoretical model for the non-centrosomal MT cytoskeleton in *Drosophila* oocytes, in which *bicoid* and *oskar* mRNAs become localised to establish the anterior-posterior body axis. Constrained by experimental measurements, the model shows that a simple gradient of cortical MT nucleation is sufficient to reproduce the observed MT distribution, cytoplasmic flow patterns and localisation of *oskar* and naive *bicoid* mRNAs. Our simulations exclude a major role for cytoplasmic flows in localisation and reveal an organisation of the MT cytoskeleton that is more ordered than previously thought. Furthermore, modulating cortical MT nucleation induces a bifurcation in cytoskeletal organisation that accounts for the phenotypes of polarity mutants. Thus, our three-dimensional model explains many features of the MT network and highlights the importance of differential cortical MT nucleation for axis formation.

*For correspondence: R.E. Goldstein@damtp.cam.ac.uk

**Competing interests:** The authors declare that no competing interests exist.

## Introduction

Microtubules (MTs) are polar cytoskeletal filaments that can adopt different global network organisations to fulfil different functions. The vast majority of studies have focussed on understanding the architecture and function of MT arrays organised by centrosomes, such as radial arrays or the mitotic spindle. In contrast, much less is known about the organisation, assembly and function of non-centrosomal MT arrays despite their ubiquity in differentiated cell types, such as neurons, epithelia, fission yeast and plants (*Bartolini and Gundersen, 2006*; *Carazo-Salas and Nurse, 2006*). In a variety of cell-types, non-centrosomal MT arrays play an essential role in directing the subcellular localisation of mRNAs to spatio-temporally control gene expression. In neuronal dendrites, for example, MTs form bidirectional arrays with MT plus-ends both pointing away and towards the cell body (*Conde and Caceres, 2009*; *Kapitein and Hoogenraad, 2011*). MTs and associated motor proteins were implicated in the transport of mRNAs along dendrites (*Bramham and Wells, 2007*), where their activity-dependent translation contributes to long term changes in synaptic function, neuronal circuitry and memory (*Miller et al., 2002*; *Sutton and Schuman, 2006*; *Holt and Schuman, 2013*). Similar overlapping, bidirectional MT arrays in the *Xenopus* oocyte have been proposed to mediate the transport of Vg1 mRNA to the vegetal cortex, where it orchestrates germ layer patterning (*Messitt et al., 2008*; *Gagnon et al., 2013*). The *Drosophila* oocyte is probably the best-studied

**eLife digest** Cells contain a network of filaments known as microtubules that serve as tracks along which proteins and other materials can be moved from one location to another. For example, molecules called messenger ribonucleic acids (or mRNAs for short) are made in the nucleus and are then moved to various locations around the cell. Each mRNA molecule encodes the instructions needed to make a particular protein and the network of microtubules allows these molecules to be directed to wherever these proteins are needed.

In female fruit flies, an mRNA called *bicoid* is moved to one end (called the anterior end) of a developing egg cell, while another mRNA called *oskar* is moved to the opposite (posterior) end. These mRNAs determine which ends of the cell will give rise to the head and the abdomen if the egg is fertilized. The microtubules start to form at sites near the inner face of the membrane that surrounds the cell, known as the cortex. From there, the microtubules grow towards the interior of the egg cell. However, it is not clear how this allows *bicoid*, *oskar* and other mRNAs to be moved to the correct locations.

Khuc Trong et al. used a combination of computational and experimental techniques to develop a model of how microtubules form in the egg cells of fruit flies. The model produces a very similar arrangement of microtubules as observed in living cells and can reproduce the patterns of *bicoid* and *oskar* RNA movements.

This study suggests that microtubules are more highly organised than previously thought. Furthermore, Khuc Trong et al.'s findings indicate that the anchoring of microtubules in the cortex is sufficient to direct *bicoid* and *oskar* RNAs to the opposite ends of the cell. The next challenge will be to find out how the microtubules are linked to the cortex and how this is regulated.

example of mRNA transport along non-centrosomal MTs. In this system, a diffuse gradient of MTs of mixed polarity is required for the localisation of *bicoid* and *oskar* mRNAs to opposite ends of the cell (*Becalska and Gavis, 2009*). Despite the large amount of work, however, the organisation of the non-centrosomal MT cytoskeleton underlying this mRNA localisation is controversial (*MacDougall et al., 2003*; *Januschke et al., 2006*; *Zimyanin et al., 2008*), and its assembly and function are not understood. In stage 9 oocytes, MTs grow from most parts of the cell cortex into the volume (*Theurkauf et al., 1992*; *Parton et al., 2011*) thereby giving rise to a complex, three-dimensional MT network without pronounced polarity along the anterior-posterior (AP) axis (*Theurkauf et al., 1992*; *Cha et al., 2001*). In contrast to this apparent disordered organisation, *bicoid* and *oskar* mRNAs become reliably localised by Dynein and Kinesin to the anterior corners and to the posterior pole of the oocyte, respectively, thereby defining the AP axis (*Brendza et al., 2000*; *Duncan and Warrior, 2002*; *Januschke et al., 2002*; *Weil et al., 2006*; *Zimyanin et al., 2008*).

The most pronounced feature of the MT cytoskeleton at stage 9 is a gradient of cortical MTs from the anterior to the posterior pole, where MT nucleation is suppressed by the polarity protein PAR-1 (*Doerflinger et al., 2006*; *Roth and Lynch, 2009*). Live imaging of *oskar* mRNAs and direct measurements of growing MTs showed that the MT network is mostly disordered with only a weak statistical bias of about 8% more plus ends pointing towards the posterior pole (*Zimyanin et al., 2008*). This bias vanishes in absence of PAR-1 when MTs nucleate from all over the cortex (*Parton et al., 2011*). While these findings established a directionality of the MT meshwork for the first time, several questions remain unanswered. For example, mRNA transport and localisation is highly reproducible, raising doubts about whether the underlying cytoskeletal organisation can be mostly disordered. Moreover, Dynein-dependent transport to MT minus ends localises injected, so-called naive *bicoid* mRNA to the cortex closest to the injection site (*Cha et al., 2001*), which is difficult to reconcile with a cytoskeleton that is simply biased towards the posterior everywhere. Similarly, the different behaviour of so-called conditioned *bicoid* mRNA, which localises specifically to the anterior cortex irrespective of the injection site (*Cha et al., 2001*), has remained unexplained. Finally, motor-driven cytoskeletal transport is not the only transport mechanism in oocytes. Kinesin moves only 13% of *oskar* mRNA at any given time, and the remaining 87% is subject to diffusion and slow cytoplasmic flows that are driven indirectly by Kinesin activity on the MT network (*Zimyanin et al., 2008*). This raises the question if cytoskeletal transport alone is sufficient to account for mRNA localisation

(*Glotzer et al., 1997*; *Forrest and Gavis, 2003*) and highlights the importance of distinguishing its contribution to localisation from the contribution of flows and diffusion (*Serbus et al., 2005*).

Here, we present the first theoretical model for stage 9 *Drosophila* oocytes. Based on the distribution of MT nucleation sites around the cortex, this model accurately reproduces the observed distribution of MTs and cytoplasmic flows in the oocyte. It reveals that the MT cytoskeleton is compartmentalised and more ordered than previously thought. By modelling the movement of mRNAs on this network, we also show that this MT organisation is sufficient to explain the localisation of *oskar* and naive *bicoid* mRNA. Finally, we show that modulation of MT nucleation gradients causes a bifurcation in cytoskeletal organisation that explains mutant phenotypes. Thus, our results explain many features of the assembly, organisation and function of the non-centrosomal MT array in *Drosophila* oocytes and highlight the key role of differential MT nucleation or anchoring at the cortex (*Lüders and Stearns, 2007*).

## Results

MTs in the oocyte are nucleated or anchored at the cortex (*Theurkauf et al., 1992*; *Parton et al., 2011*) and grow from the membrane into the volume. The cortical MT density follows a steep gradient along the posterior-lateral cortex from high density at the anterior corners to low densities at the posterior pole (*Figure 1D*, 'Materials and methods' MT nucleation probability). In our model, we emulated these features by selecting seeding points for MTs at random positions along the oocyte cortex, comprised of two parabolic, rotationally symmetric caps that capture the typical shape of a stage 9 *Drosophila* oocyte (*Figure 1A*, 'Materials and methods' Coordinates and oocyte geometries). The density of seeding points decreases steeply along the posterior-lateral cortex to zero at the posterior pole, and decreases weakly along the anterior cortex towards the anterior centre (*Figure 1A,B*, 'Materials and methods' MT nucleation probability). Each seeding point nucleates a MT polymer that grows until it either hits a boundary or reaches a target length imposed by the aging of MTs before catastrophe (*Gardner et al., 2011*) ('Materials and methods' MT growth). In total, we computed more than 55,000 MTs for each realisation of a 3D wild-type cytoskeleton.

A cross section through the computed MT meshwork (*Figure 1A,B*) bears a striking resemblance to confocal images of Tau- (*Micklem et al., 1997*; *Ganguly et al., 2012*) and EB-1- (*Parton et al., 2011*) tagged MTs, correctly showing the pronounced AP gradient of MT density. Taking into account the orientations of all MT segments in the 3D volume, it also reproduces the experimentally measured directional bias (*Parton et al., 2011*) with 8.5% more MT segments pointing posteriorly than anteriorly (*Figure 1A*, inset). The directional bias in a 2D slice varies depending on the depth of the slice in the oocyte, ranging from 50.7% of posteriorly oriented MT segments at the cortex to 60.6% in the mid plane (*Figure 1B*, inset). This demonstrates that measurements in 2D confocal slices poorly characterise the fully extended 3D system.

Central to understanding mRNA localisation is the question of MT orientations. Calculating the local vectorial sum of MT segments on a coarse-grained grid gives the local MT orientations in the computed cytoskeleton. Local orientations of an individual realisation of the cytoskeleton (*Figure 1A,B*) show poor global network order (*Figure 1C*). However, cargo localisation in the oocyte does not involve only a single cytoskeletal realisation. MTs disappear after 5–10 min in the presence of Colchicine (*Theurkauf et al., 1992*; *Zhao et al., 2012*) (V Trovisco, personal communication), a drug that blocks MT growth and destabilizes dynamic MTs, indicating that the whole network turns over within minutes. Thus, the oocyte samples many tens to a hundred of independent MT organisations over the 6–9 hr of stage 9. Summation of local orientations over an ensemble of 50 independent realisations of the computed MT network reveals a striking spatial partitioning of the cytoskeleton into several subcompartments (*Figure 1F*). At the anterior, the mean orientation points posteriorly, while MTs at the lateral sides on average point inwards toward the AP-axis. Counting the number of MT segments that contribute to each grid box in the ensemble also shows the distribution of MT density (*Figure 1H*). In agreement with experiments (*Figure 1D*), the MT density exhibits a pronounced AP gradient with highest values in the anterior corners, even when MT seeding is uniform on the anterior surface.

In the limit of a large ensemble, the MT polymers sample all possible initial directions and possible lengths at every point along the cortex. In this limit, we can test the predicted compartmentalised MT organisation by constructing a second model in which MTs are represented as straight rods. The net

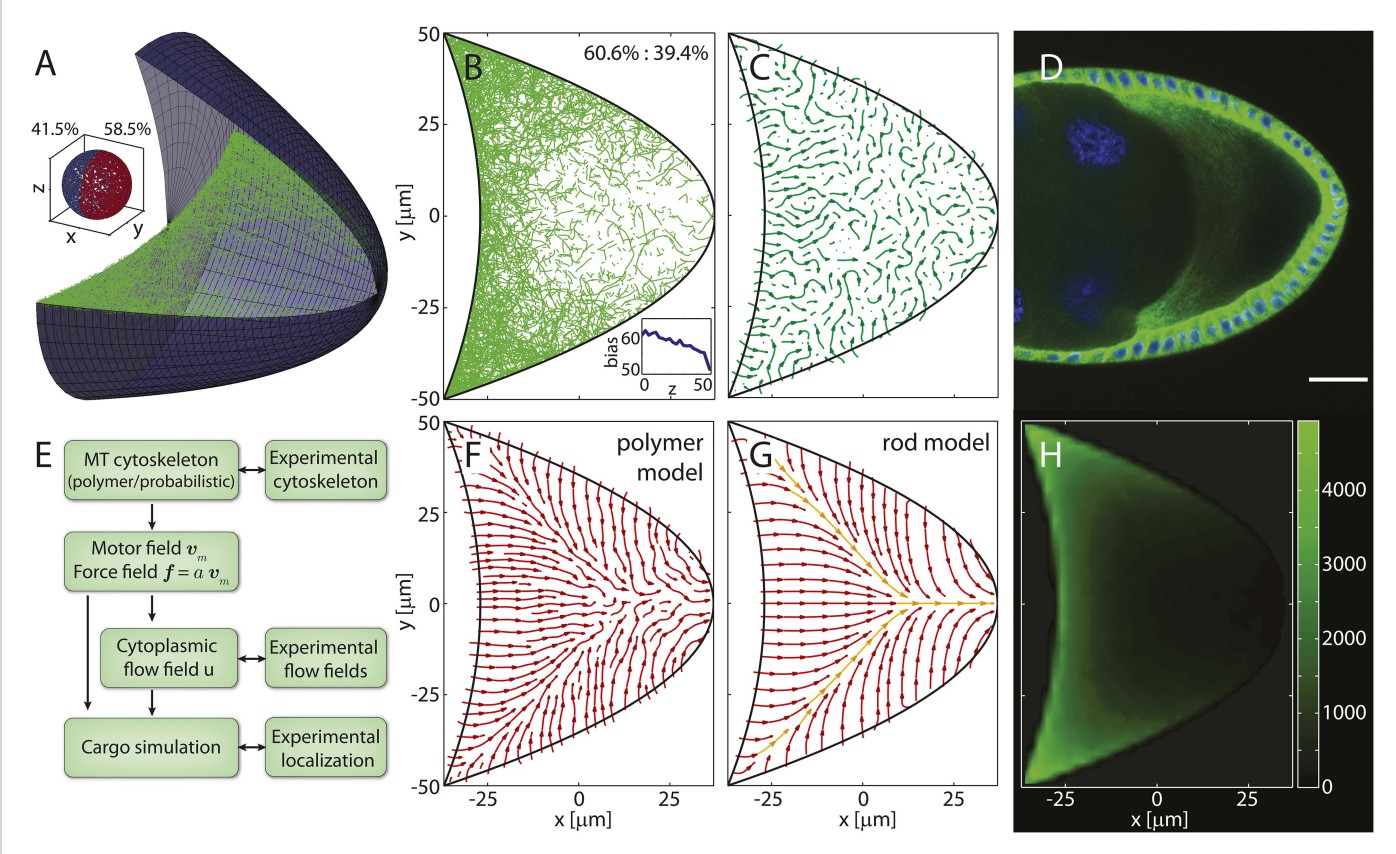

**Figure 1**. Models for the microtubule (MT) meshwork show local order in the cytoskeleton. (**A**) 3D geometry of a stage 9 Drosophila oocyte (grey, anterior to the left, posterior to the right) containing more than 55,000 MT seeding points. From the corners to the centre, MT seeding density decreases weakly along the anterior ($k_A = 1000$ μm, $h_0^A = 0.8$) and strongly along the posterior-lateral cortex ($k_P = 150$ μm, $h_0^P = 0$, see 'Materials and methods' MT nucleation probability). Nucleated MT polymers are stiff random walks, initially pointing in a random direction. Only MT segments in a cross section are shown (green) to emulate confocal images. MT target lengths are chosen from a probability distribution that accounts for the MT aging process. The mean target length is set to a fraction $\epsilon$ of the anterior-posterior (AP) axis length, here $\epsilon = 0.5$ ('Material and methods' MT growth). The inset shows the 3D angular distribution of 0.5% of all MT segments with 3D statistical bias. (**B**) Cross section through the MT cytoskeleton shown in **A** with 2D directional bias (top right). The inset shows 2D posterior bias (in percent) as function of depth (bottom right). (**C**) Local vector sum of MT segments from the cross section in panel **B** on a coarse-grained grid shown as streamlines that visualize local directionality. (**D**) Staining of $\alpha$-tubulin (green) shows MT density distribution in a fixed stage 9 oocyte. Nuclei in blue (DAPI), scale bar is 30 μm. (**E**) Schematic detailing the work flow in the model and comparisons to experiments. (**F**) Local directionality of MT cross section as in panel **C** for an average over 50 independent realizations of the cytoskeleton. (**G**) Local directionality computed from the rod model with the same parameters as in panels **A**, **B** but shortened MT lengths ('Material and methods' Motor velocity field). Orange arrows show the separatrices between subcompartments. (**H**) MT density distribution computed from 50 realizations of the polymer model.

The following figure supplement is available for figure 1:

**Figure supplement 1**. Compartmentalization of the MT cytoskeleton is robust to changes in oocyte geometry.

orientation of the cytoskeleton at a given point inside the oocyte is then computed as weighted sum of MT contributions from each point along the boundary ('Materials and methods' Model setup).

With a shorter MT target length to compensate for the effectively longer length of straight rods compared to curved polymers, computation of net MT orientations in the rod model shows a topology (*Figure 1G*) that confirms the mean topology in the polymer model (*Figure 1F*). The three compartments of a posterior-pointing anterior section and two inwards-pointing lateral sections are effectively bounded by separatrices (*Figure 1G*, orange arrows). Cargo molecules that are transported on MTs to their plus ends move along the arrows and converge at the separatrices, eventually leading to the posterior pole as the sole point of attraction in the entire oocyte volume (the attractor). By contrast, cargo that is transported to MT minus ends moves opposite to the arrows and diverges away from the separatrices.

This mean topology of the MT cytoskeleton is insensitive to the exact choice of the MT nucleation probability and to the choice of the MT length distribution ('Materials and methods' MT nucleation probability, MT growth). It also remains unchanged in a differently shaped oocyte geometry in both the polymer model and the rod model (*Figure 1—figure supplement 1*). In summary, both models show that even disordered non-centrosomal MT arrays can feature well-defined mean organisations, and that the MT cytoskeleton in oocytes is organised in a compartmental fashion.

The suitably scaled local vectorial sum of MT segments (*Figure 1C*) for an individual realisation of the polymer model (*Figure 1A,B*) is a vector field $v_m$ which represents active Kinesin-driven transport on the cytoskeleton ('Materials and methods' Motor velocity field). In vivo during stage 9, the oocyte cytoplasm undergoes slow cytoplasmic flows that are abolished in kinesin heavy chain mutants. This indicates that flows are driven by kinesin-dependent transport of an unknown cargo through the viscous cytoplasm (*Palacios and St Johnston, 2002*; *Serbus et al., 2005*), thus making cytoplasmic flows a secondary read-out of cytoskeletal organisation (*Khuc Trong et al., 2012*). Therefore, we next tested if our computed polymer MT cytoskeleton can produce flows that are consistent with observed cytoplasmic streaming.

Measurements of speeds of autofluorescent yolk granules in live stage 9 oocytes showed that mean flow velocities are slow (*Figure 2F*). The physics of slow incompressible fluid flows $u$ driven by forces $f$ is described by the Stokes equations

$$0 = -\nabla p + \mu \nabla^2 u + f \ , \nabla \cdot u = 0 \ . \tag{1}$$

We make the simplest possible assumption that the forces $f$ are proportional to the motor-velocity field, and use experimentally measured flow speeds to calibrate the scalar factor of proportionality. By solving the Stokes equations, we then computed the full 3D fluid flow field (*Figure 2A*) corresponding to an individual realisation of the MT cytoskeleton (*Figure 1A*), and 2D cross sections through the 3D field (*Figure 2B*) were compared to in vivo flow patterns visualised by particle image velocimetry (PIV, *Figure 2C*).

Despite large variability, both computed and in vivo flows show very similar patterns, generally being strongest in the anterior half of the oocyte, and weaker in the posterior half (*Figure 2B,C*). Computed flows occasionally reach further into the posterior half than typically seen in PIV flow fields, thereby slightly overestimating the range of flows. However, except in rare cases, computed flows do not reach the posterior pole. This result is largely independent of the presence of the oocyte nucleus which occupies less than 2.5% of the oocyte volume, even for small oocytes at the beginning of stage 9 (*Figure 2—figure supplement 1*, 'Materials and methods' Nucleus to oocyte volume ratio, Impact on flow field). Thus, computed flows appear consistent with observed slow cytoplasmic streaming.

We tested next if our computed cytoskeleton in combination with the derived cytoplasmic flows and diffusion can account for dynamic mRNA transport and localisation. We described the mRNA distributions as continuous concentration fields. *oskar* mRNA is assumed to reside in either one of two states: the Kinesin-bound state with concentration $c_b$, in which cargo is transported actively on the cytoskeleton-derived motor-velocity field $v_m$ (*Figure 1C*), or the unbound state with concentration $c_u$, in which cargo is transported by the cytoplasmic flows $u$ (*Figure 2A*) and diffuses with diffusion constant $D$. Cargo can exchange between both states by chemical reactions, thereby resulting in the reaction-advection-diffusion equations:

$$\partial_t \, c_b + \nabla \cdot (v_m \, c_b) = k_b \, c_u - k_u \, c_b, \tag{2}$$

$$\partial_t \, c_u + \nabla \cdot (u \, c_u) = -k_b \, c_u + k_u \, c_b + D\nabla^2 c_u.$$

Parameter values were constrained with experimentally measured values ('Materials and methods' Parameter values). Furthermore, throughout the simulation we cycled through the pairs of fluid flow $u$ and motor-velocity fields $v_m$ (*Figure 3*) to account for the dynamic nature of the MT network (*Parton et al., 2011*) and flow patterns, which change over time scales of minutes compared to the 6–9 hr during which *oskar* mRNA is localised ('Materials and methods' Autocorrelation function).

Starting from an initial diffuse cloud of *oskar* mRNA in the centre of the oocyte (*Figure 3A,D,E*), simulations show that the mRNA quickly concentrates in the centre before forming a channel from the centre of the oocyte towards the posterior pole (*Figure 3E*). Formation of this channel reflects the

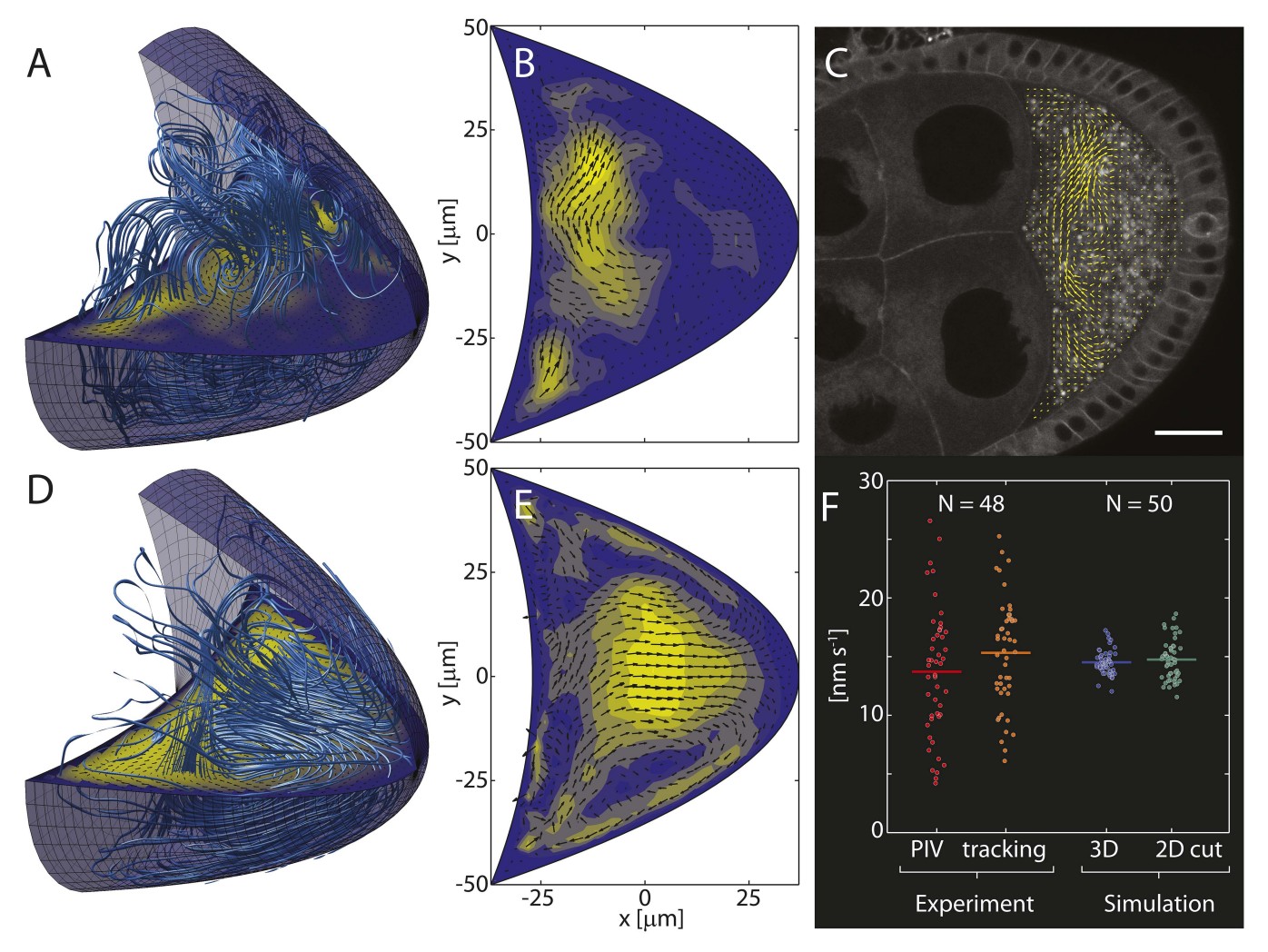

**Figure 2**. Computed cytoplasmic flow fields capture key elements of in vivo flows. (**A**) Streamlines (light blue lines) visualize the 3D cytoplasmic flow field computed from the realization of the cytoskeleton shown in **Figure 1A**. The horizontal plane shows a 2D cross-section through the 3D field. Anterior to the left, posterior to the right. (**B**) Cross-section through the 3D field shown in panel **A** with arrows indicating flow directions and colouring indicating flow speeds. (**C**) Confocal image of a live stage 9 oocyte. Arrows show the flow field computed from particle image velocimetry (PIV) of streaming yolk granules and averaged over ≈5 min. Scale bar is 25 μm. (**D**) Same as **A**, but showing the mean flow organization for an average of 100 individual 3D flow fields, analogous to the mean organization of the MT cytoskeleton in **Figure 1F**. (**E**) Same as **B** for the average in panel **D**. (**F**) Mean fluid flow speeds were obtained by PIV (red, 13.7 ± 0.8 nm/s, mean ± sem) and automatic particle tracking (orange, 15.3 ± 0.7 nm/s, mean ± sem) from 48 oocytes. Experimentally measured flow speeds were used to calibrate the forces f in the Stokes **Equation 2** such that the computed mean speeds in 3D (blue, mean: 14.5 nm/s) or in 2D cross sections (green, mean: 14.8 nm/s) match the measured values. The larger spread in experimental flow speeds may reflect greater variability of motor activity, cytoplasmic composition, geometry or age in vivo.

The following figure supplement is available for figure 2:

**Figure supplement 1**. The oocyte nucleus covers a negligible fraction of the oocyte volume and disturbs the flow field only locally.

locally inwards orientation of MTs in the posterior half of the oocyte (**Figure 1E,F**). This transient state closely resembles transient patterns of fluorescently-tagged *oskar* mRNA during the transition between stages 8 and 9 (**Figure 3B**), thereby supporting the notion that our computed MT cytoskeleton correctly captures key aspects of the in vivo MT network.

Upon reaching the posterior cortex, *oskar* mRNA is translated to produce Long Osk (**Vanzo and Ephrussi, 2002**), which is involved in anchoring *oskar* mRNA. However, *oskar* mRNA localises normally

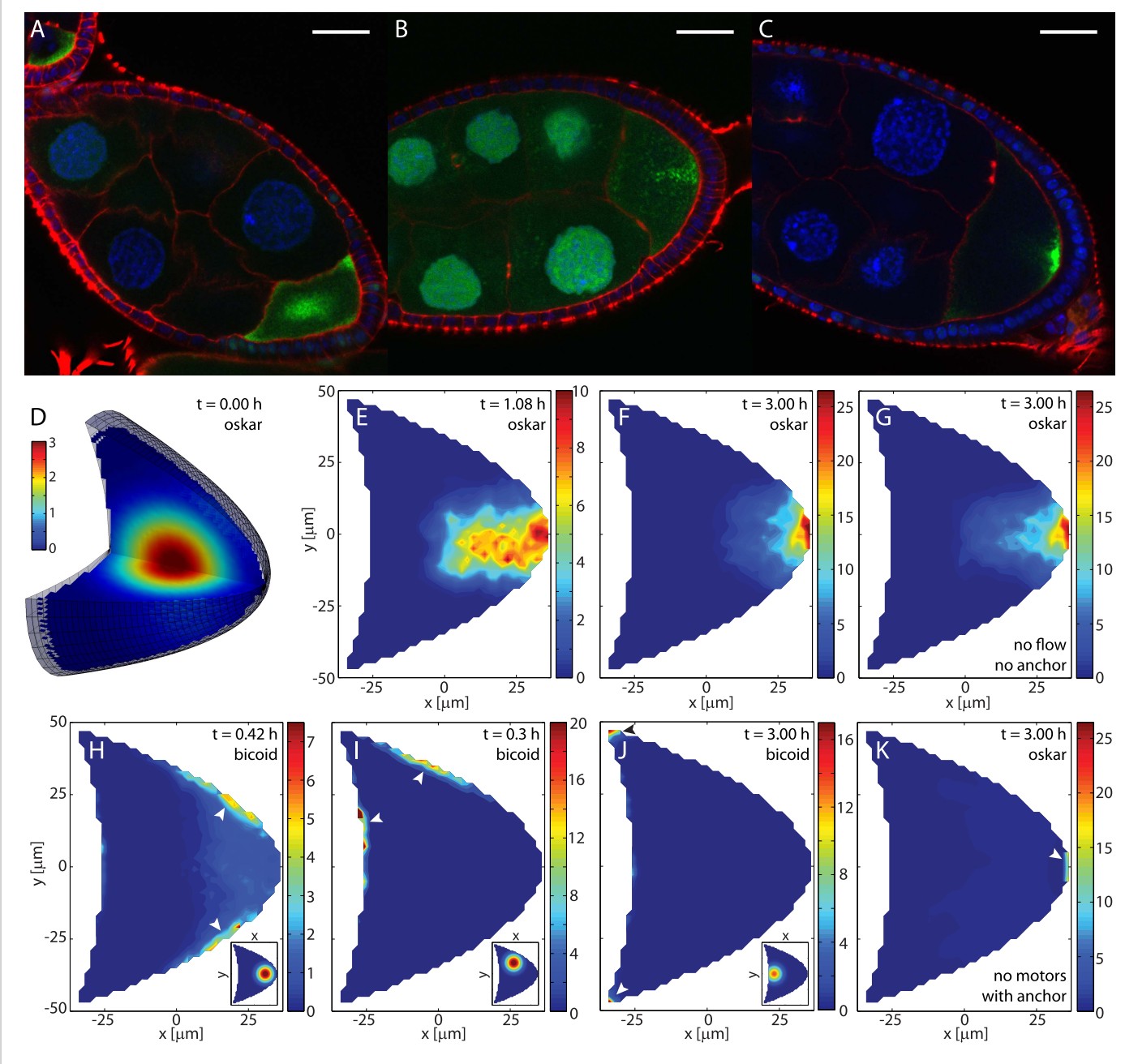

**Figure 3**. The model recapitulates *oskar* and *bicoid* mRNA transport, implying dominance of cytoskeletal transport. (**A–C**) Shown are fixed oocytes with *oskar* MS2 GFP (green) and stained with DAPI (blue) and Phalloidin (red). *oskar* mRNA forms a central cloud at late stage 8 (**A**), a collimated channel while moving to the posterior (**B**), and a posterior crescent at stage 9 (**C**). Scale bars are 25 μm. (**D**) 3D oocyte showing the initial *oskar* mRNA cargo distribution. (**E**, **F**) Cross sections through a simulation of *oskar* mRNA transport with diffusion, motor-transport and flows showing the distribution of total cargo $c_b + c_u$ at the indicated time points. No posterior anchor is present. Simulations of three 6 hr cycles through 50 ($v_m$,$u$)-pairs in random order once (twice, see *Figure 4*). For simulations of 1.5 hr, 25 ($v_m$,$u$)-pairs were chosen at random. Compare to experimental observations in panels **B** and **C**. (**G**) Same simulation as in **D–F**, but without cytoplasmic flows, showing largely identical localisation as in **F**. (**H**) Simulation of *bicoid* mRNA transport shows the mRNA quickly accumulating at the nearest cortex (arrowheads) when injected in the posterior (inset), corresponding to the behavior of naive *bicoid*. (**I**) Same as in **H**, corresponding to naive *bicoid* mRNA injection at the anterior-dorsal region (inset). (**J**) Same as in **H** for injection at the anterior middle (inset), showing that over long times simulated *bicoid* mRNA localises to the anterior corners (arrowheads). Localization to the anterior depends on sufficient proximity of the injection site as observed for injections of naive *bicoid* mRNA. (**K**) Same simulation for *oskar* mRNA as in **D–F**, but with a posterior anchor (arrowhead) and without active motor-driven transport. Compare to panels **F** and **G**.

at stage 9 when anchoring is disrupted (*Micklem et al., 2000*; *Vanzo and Ephrussi, 2002*). Anchoring is therefore not necessary at stage 9, and we did not include it in our model at this point. Despite the lack of any anchoring, *oskar* mRNA forms a posterior crescent after 1.5 hr and becomes highly concentrated by the end of the simulation (*Figure 3F*), correctly capturing observations of in vivo localisation (*Figure 3C*). A 4 μm thick slice at the posterior pole contains 12.5% of total cargo in the oocyte (*Figure 3F*), showing that continual transport combined with slow diffusion is sufficient to reach and maintain high concentrations of mRNA.

In our description of transport (*Equation 2*), cargo acts as an unspecific passive tracer. Passive tracers merely follow the transport fields $v_m$ and $u$, and these two fields must contain all information about the localisation sites. We therefore asked which field contains most information about localisation. We first tested localisation in the absence of cytoplasmic flows by setting the fluid flow field identical to zero $u \equiv 0$. This situation is similar to but more severe than in slow Kinesin mutants (*Serbus et al., 2005*). Starting from the central *oskar* mRNA cloud (*Figure 3D*), we found that *oskar* mRNA still localises to and forms a concentrated crescent at the posterior pole (*Figure 3G*). A 4 μm thick slice off the posterior pole contains 11.3% of total available *oskar* mRNA cargo, only marginally less than localisation with cytoplasmic flows present. This suggests that the cytoskeletal transport alone localises the majority of mRNA. Cargo localisation in the absence of cytoskeletal transport can be tested by either setting the motor-velocity field identical to zero $v_m \equiv 0$, or by setting the binding constant $k_b$ to zero. In either case, cargo disperses throughout the oocyte without forming a central channel and completely fails to produce any posterior crescent. We further asked whether the flow could localise cargo to wild-type levels if an anchor captured cargo at the posterior. To this end, we replaced the bound state in the model by an anchored state to which cargo can bind only at the very posterior boundary. Binding to the anchor occurs with a reaction rate constant that is 10 times larger than the reaction rate constant $k_b$ used for *oskar* mRNA binding to the MT cytoskeleton ('Materials and methods' Parameter values). Moreover, anchored cargo cannot be released, thus making this anchor a perfect sink with stronger trapping properties than any realistic anchor. Under these idealised conditions, *oskar* mRNA accumulates slightly at the anchor (*Figure 3K*). However, a 4 μm thick posterior slice contains only 2.7% of total cargo in the oocyte, about 78% less cargo than in the wild-type simulations, and only 25% more cargo compared to the purely diffusive case, which localises 2.2% of cargo. Thus, even under the most favourable conditions, the cytoplasmic flows cannot localise mRNA to wild-type levels, arguing against a mixing and entrapment mechanism for *oskar* mRNA localisation at stage 9. Despite the small fraction of actively transported cargo (13%), motor-driven mRNA transport is both necessary and sufficient to account for *oskar* localisation.

In contrast to *oskar* mRNA, *bicoid* mRNA is believed to be transported by Dynein (*Duncan and Warrior, 2002*; *Januschke et al., 2002*; *Weil et al., 2006*), but other parameters such as the fraction of bound cargo and the active transport speeds are similar to *oskar* mRNA (*Belaya, 2008*) (V Trovisco, personal communication). To test whether the proposed cytoskeletal organisation also captures transport and localisation of injected *bicoid* mRNA, we inverted the directions of the motor-velocity fields (*Figure 1C*) to account for the minus end directed transport by Dynein instead of the plus end directed transport by Kinesin. Other parameters including cytoplasmic flow fields remained unchanged.

Simulations show that *bicoid* mRNA accumulates at both posterior-lateral sides (*Figure 3H*) when initially placed in the posterior half of the oocyte (*Figure 3H*, inset). This is in very good agreement with experimental observations when naive *bicoid* mRNA is injected into this posterior region (*Cha et al., 2001*). When *bicoid* mRNA is placed initially in the anterior-ventral region (*Figure 3I*, inset), it accumulates at the anterior and lateral cortex (*Figure 3I*), again in concordance with the experimental observations (*Cha et al., 2001*). This simulated localisation of *bicoid* mRNA remains virtually identical in the absence of cytoplasmic streaming. Thus, splitting of a bulk amount of injected *bicoid* mRNA occurs when the RNA is placed on the border (separatrices) between two diverging subcompartments of the MT cytoskeleton, each one transporting part of the cloud towards the adjacent cortex. The agreement between simulations and experiments therefore further supports an on average compartmentalised MT cytoskeleton, and naive *bicoid* mRNA, like *oskar* mRNA, unspecifically traces out the MT cytoskeleton.

In contrast to naive *bicoid* mRNA, endogenous *bicoid* mRNA is transported into the oocyte after being transcribed in neighbouring nurse cells where it also becomes modified in presence of the protein Exuperantia. Endogenous *bicoid* mRNA can be approximated by so-called conditioned *bicoid* mRNA that is first injected into the nurse cells, then sucked out and injected into the oocyte.

Conditioned *bicoid* mRNA localises specifically to the anterior surface within 30 min, even if this surface is not closest to the injection site (*Cha et al., 2001*). In the model, starting from an initial distribution at the anterior surface where endogenous *bicoid* mRNA enters the oocyte, cargo quickly moves to the anterior cortex before concentrating in the anterior corners after several hours of simulated time (*Figure 3J*). This resembles the anterior ring-like localisation of *bicoid* mRNA in wild-type stage 9 oocytes (*St Johnston et al., 1989*; *Weil et al., 2006*), and occurs independently of cytoplasmic flows. However, the model does not reproduce transport specifically to the anterior surface when injection is further away from the anterior. This suggests that Exuperantia activity somehow either enables *bicoid* mRNA to move along an unidentified population of MTs (*Cha et al., 2001*; *MacDougall et al., 2003*) that are not included in the computed cytoskeleton, or that it allows localisation of the mRNA by an unknown MT-independent mechanism.

We have shown so far that cortical gradients of MT nucleation are sufficient to assemble a functional, compartmentalised MT cytoskeleton that successfully localises *oskar* mRNA and naive *bicoid* mRNA. Next, we asked if cortical MT nucleation can also produce cytoskeletal organisations that explain mutants with partial or complete polarity defects. Mutations interfering with the posterior follicle cells that surround the oocyte, for example, can disrupt the proper positioning of mRNAs (*Gonzalez-Reyes et al., 1995*; *Roth et al., 1995*). Specifically, follicle cell clones (ras$^{\Delta C40b}$) that are adjacent to only one side of the oocyte posterior repel cargo from that side of the cortex (termed clone adjacent mislocalisation [*Poulton and Deng, 2006*]), likely due to an altered MT organisation. To mimic this phenotype, we used an ensemble of wild-type cytoskeletons and artificially added MT nucleation sites to a patch on one side of the posterior pole. Simulations of cargo transport with diffusion, motor transport and flows show that *oskar* mRNA is repelled from this patch and localises to the adjacent posterior boundary, in agreement with experiments (*Figure 4—figure supplement 2*).

Mislocalisation of mRNAs also occurs in mutants of the polarity protein PAR-1, which acts to suppress MT nucleation at the posterior pole of the oocyte (*Shulman et al., 2000*; *Doerflinger et al., 2010*). In strong *par-1* hypomorphs with strongly reduced PAR-1 expression, MT minus ends occupy the whole cortex including the posterior pole (*Parton et al., 2011*), thereby causing *oskar* mRNA to mislocalise to a dot in the centre of almost 90% of mutant oocytes (*Doerflinger et al., 2006*). To test this phenotype in the model we distributed MT seeding points uniformly on the posterior surface (*Figure 4G*, inset) while keeping the seeding density on the anterior surface constant. The AP gradient of MT density then vanishes and the directional bias of MT segments evens out (*Figure 4G*, ensemble-averaged 3D-bias: 50.1%:49.9%). The ensemble-averaged local orientations of MTs show that MTs point towards a single focal point in the centre of the oocyte (*Figure 4G*) that acts as a stable fixed point of the system. Computation of 3D flow fields for each realisation in the new ensemble (3D mean: 13.5 nm/s; 2D mean: 14.5 nm/s, N = 50) and simulations of transport with and without cytoplasmic flows show *oskar* mRNA concentrating in a cloud in the centre of the oocyte (*Figure 4C*). *oskar* mRNA accumulation to a central dot also remains unchanged if the posterior MT seeding density is decreased slightly (*Figure 4H*, inset) as might be expected for hypomorphs that do not abolish PAR-1 expression completely.

Interestingly, *bicoid* mRNA simulated either with slightly decreased (*Figure 4H*) or with uniform (*Figure 4G*) posterior MT nucleation not only localises to the anterior corners as in wild-type but also enriches at the posterior pole (*Figure 4D*, arrowheads), despite injection close to the anterior (*Figure 4D*, inset). This matches experiments showing *bicoid* mRNA mislocalisation to both anterior and posterior in *gurken*, *torpedo* and *cornichon* mutants (EGFR mutants) in which posterior follicle cells fail to signal a MT reorganisation in the oocyte (*Gonzalez-Reyes et al., 1995*; *Roth et al., 1995*).

Failure of the external signal to the oocyte is thought to have the same effect as *par-1* hypomorphs, but previous experiments with *bicoid* mRNA in *par-1* hypomorphs resulted in ambiguous localisations (*Shulman et al., 2000*; *Benton et al., 2002*). Here, using confocal fluorescence in situ hybridisation, our experiments show that *bicoid* mRNA localises primarily to the anterior corners and to the posterior pole in strong *par-1* hypomorphs (*Figure 4L*), thereby mirroring EGFR mutants and agreeing with the model predictions. Thus, (near-)uniform MT nucleation along the posterior and lateral cortex is sufficient to explain the polarity phenotypes observed for *oskar* and *bicoid* mRNAs in *gurken*, *torpedo* and *cornichon* mutants and in strong *par-1* hypomorphs.

In 10% of strong *par-1* hypomorphs and 30% of weak *par-1* hypomorphs with weakly reduced PAR-1 expression, *oskar* mRNA is only partially mislocalised, combining a wild-type-like posterior crescent with a centrally mislocalised dot (*Doerflinger et al., 2006*). We therefore modulated the

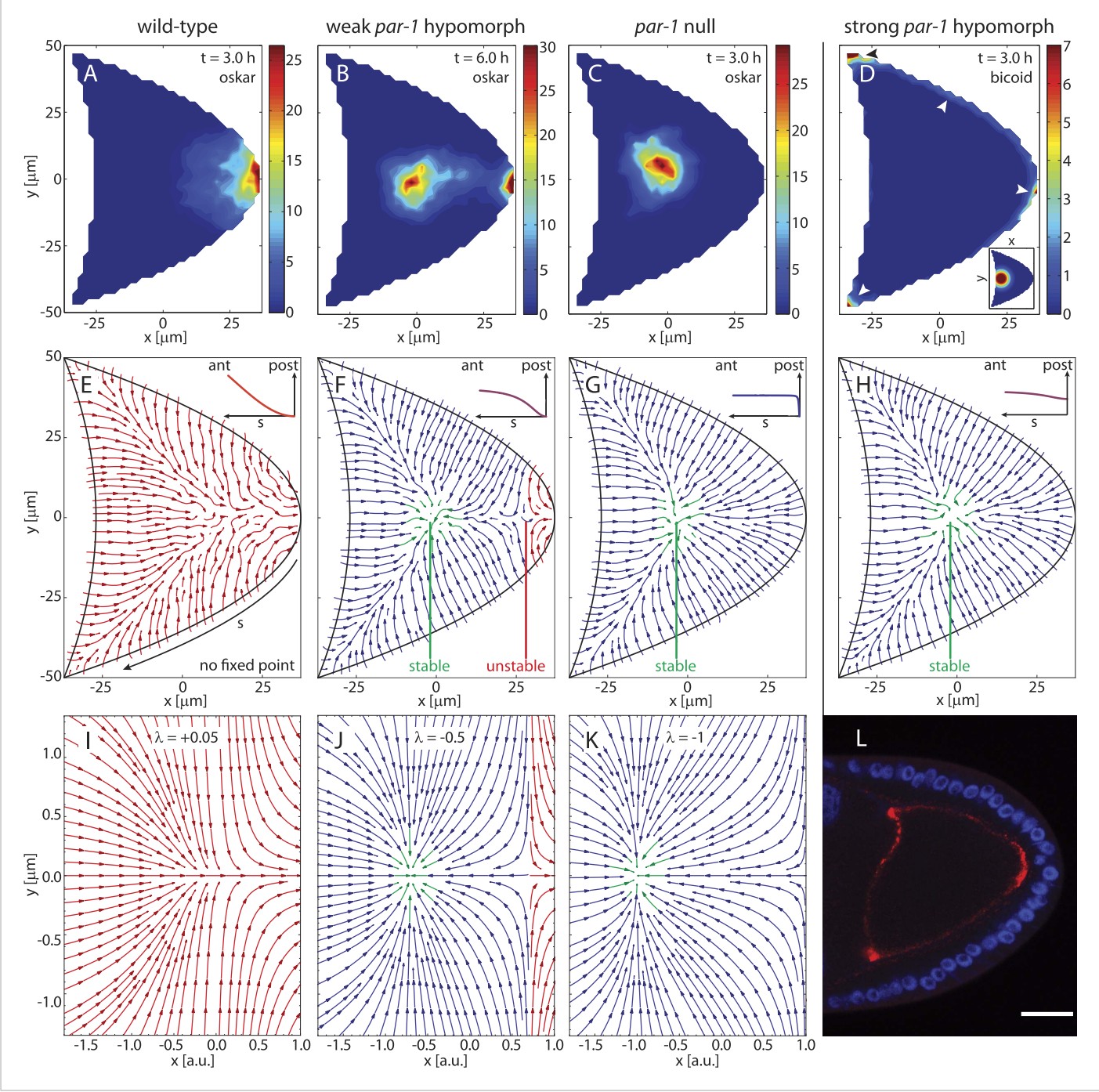

**Figure 4**. The cortical MT seeding density determines sites of mRNA localisations and gives rise to a bifurcation in the cytoskeleton. (**A–E**) Simulations of *oskar* (**A–C**) and *bicoid* (**D**) mRNA transport with diffusion, cytoskeletal transport and flow for the cytoskeletal architectures shown in panels **E–H**, and their corresponding flow fields. For *oskar* mRNA, simulations reproduce wild-type localisation (**A**, same as *Figure 3F*), and partial (**B**) or complete mislocalisation (**C**). Initial condition as in *Figure 3D*. Simulation times were occasionally increased to 6 hr (**B**) to rule out transient concentration patterns. For *bicoid* mRNA, simulations capture mislocalisation to both anterior and posterior as in gurken/torpedo/cornichon mutants and in strong *par-1* hypomorphs (**D**, arrowheads). Time points as indicated. The inset in **D** shows initial condition. (**E–H**) Average local MT orientations in ensembles of 50 realizations of the polymer model for varying MT seeding densities along arclength *s* (see panel **E**) of the posterior-lateral cortex (insets). A MT seeding density that increases from a wild-type gradient (**E**, $h_0^P = 0$, $k_P = 150$ μm) laterally towards the posterior (**F**, $h_0^P = 0$, $k_P = 17.5$ μm) to a near-uniform distribution (**G**, $h_0^P = 0$, $k_P = 1$ μm) shows a saddle-node bifurcation by creating a pair of stable (green) and unstable (red) fixed points. (**H**) A MT seeding density that is slightly lower at the posterior pole ($h_0^P = 0.7$, $k_P = 40$ μm) produces mean MT orientations virtually indistinguishable from uniform seeding density (compare to **G**). (**I–K**) Vector field of the mathematical normal form of the 2D saddle-node bifurcation ('Materials and methods' Bifurcation normal

*Figure 4. continued on next page*

*Figure 4. Continued*

form). Values of the bifurcation parameter $\lambda$ as indicated. Fixed points are located at positions $x = \pm \sqrt{\lambda}$. (**L**) Fluorescence in-situ hybridization to *bicoid* mRNA in a strong *par-1* hypomorph. The mRNA (red) localises around the cortex, with most accumulation at the anterior corners and at the posterior pole (compare to panel **D**, arrowheads). Scale bar is 25 μm.

The following figure supplements are available for figure 4:

**Figure supplement 1**. Either the MT seeding density or the MT length can act as bifurcation parameter.

**Figure supplement 2**. Lateral MT growth produces the clone-adjacent-mislocalisation phenotype (see main text).

MT nucleation along the posterior-lateral cortex to test if this is sufficient to produce intermediate cytoskeletal organisations between wild-type (*Figure 4F*) and strong *par-1* hypomorphs (*Figure 4H*).

Increasing MT nucleation laterally towards the posterior pole (*Figure 4G*, inset) again creates a stable focus in the centre of the oocyte. However, its domain of attraction does not cover the entire oocyte. Instead, a small basin of attraction towards the posterior pole persists (*Figure 4G*, red), separated from the basin of attraction of the stable focus by an unstable saddle-node point. Further increasing the seeding density towards the posterior pole shows that the pair of stable and unstable fixed points move further apart (*Figure 4—figure supplement 1*) until the unstable point no longer resides inside the oocyte geometry, thereby giving rise to the strong *par-1* hypomorph topology (*Figure 4H,I,J*). Cargo simulated on cytoskeletons with stable and unstable fixed points and their corresponding flow fields can split into two separate accumulations, first at the posterior pole and second in a dot along the AP-axis (*Figure 4B*). This agreement with experimental results suggests that intermediate levels of PAR-1 at the posterior lead to a cytoskeleton with a potential barrier between the oocyte centre and its posterior pole. It also shows that modulation of MT nucleation along the posterior-lateral cortex alone is sufficient to capture this. Because the potential barrier only affects the bound state, diffusion and flows can aid posterior localisation in this context by pushing unbound cargo across the unstable point and increasing the amount of *oskar* mRNA that reaches the posterior pole. In summary, variations of MT nucleation along the cortex not only explain mRNA localisations in wild-type but can also account for the mislocalisations of *bicoid* and *oskar* mRNAs in polarity mutants.

The creation of stable and unstable fixed points in the transition from wild-type to strong *par-1* hypomorph topology constitutes a classical saddle-node bifurcation (*Figure 4K–M*, 'Materials and methods' Bifurcation normal form) with the cortical MT seeding density as the bifurcation parameter. Interestingly, in both the polymer model and the rod model, the mean length of MTs acts as a second bifurcation parameter. For example, for sufficiently short MT filaments (*Figure 4K*), few MTs from the anterior can reach and contribute to orientations in the posterior half of the oocyte. In the posterior half, MTs from the lateral side either point towards the anterior, thereby combining with anterior MTs to create a stable node, or towards the posterior, to form a domain of attraction at the posterior pole. Therefore, for sufficiently weak contributions from anterior MTs, MTs from the lateral sides create the unstable tipping-point.

To understand more generally how the MT lengths and the posterior-lateral distribution of seeding points together influence the three different cytoskeletal topologies (*Figure 4E–G*) we computed the parameter space of the straight rod model (*Figure 5*). If MTs are absent at the posterior pole, the wild-type topology (*Figure 5Di,vi*) covers a large fraction of parameter space (*Figure 5A*, red) for sufficiently long MTs (large $\epsilon$) and cortical MT nucleation gradient (large $k_P$). The creation of two fixed points splits the oocyte into different domains of attraction (*Figure 5A*, blue, *Figure 5Di,ii*) for either shorter MTs (*Figure 5A*, vertical dashed arrow; *Figure 4—figure supplement 1I,N*) or for increasingly uniform MT nucleation (*Figure 5A*, horizontal dashed arrow; *Figure 4—figure supplement 1I,H*). For almost uniform nucleation along the entire posterior-lateral cortex, the unstable fixed point exits the geometry, leaving behind only a single stable focus (*Figure 5A*, green, arrowhead, *Figure 5Diii,iv*). If some MT nucleation is allowed at the posterior pole this region of strong *par-1* hypomorph topology expands substantially (*Figure 5B,C*, green). Interestingly, if MTs are sufficiently long, MTs from the lateral and anterior cortex can reach and overpower those from the posterior pole, thus restoring

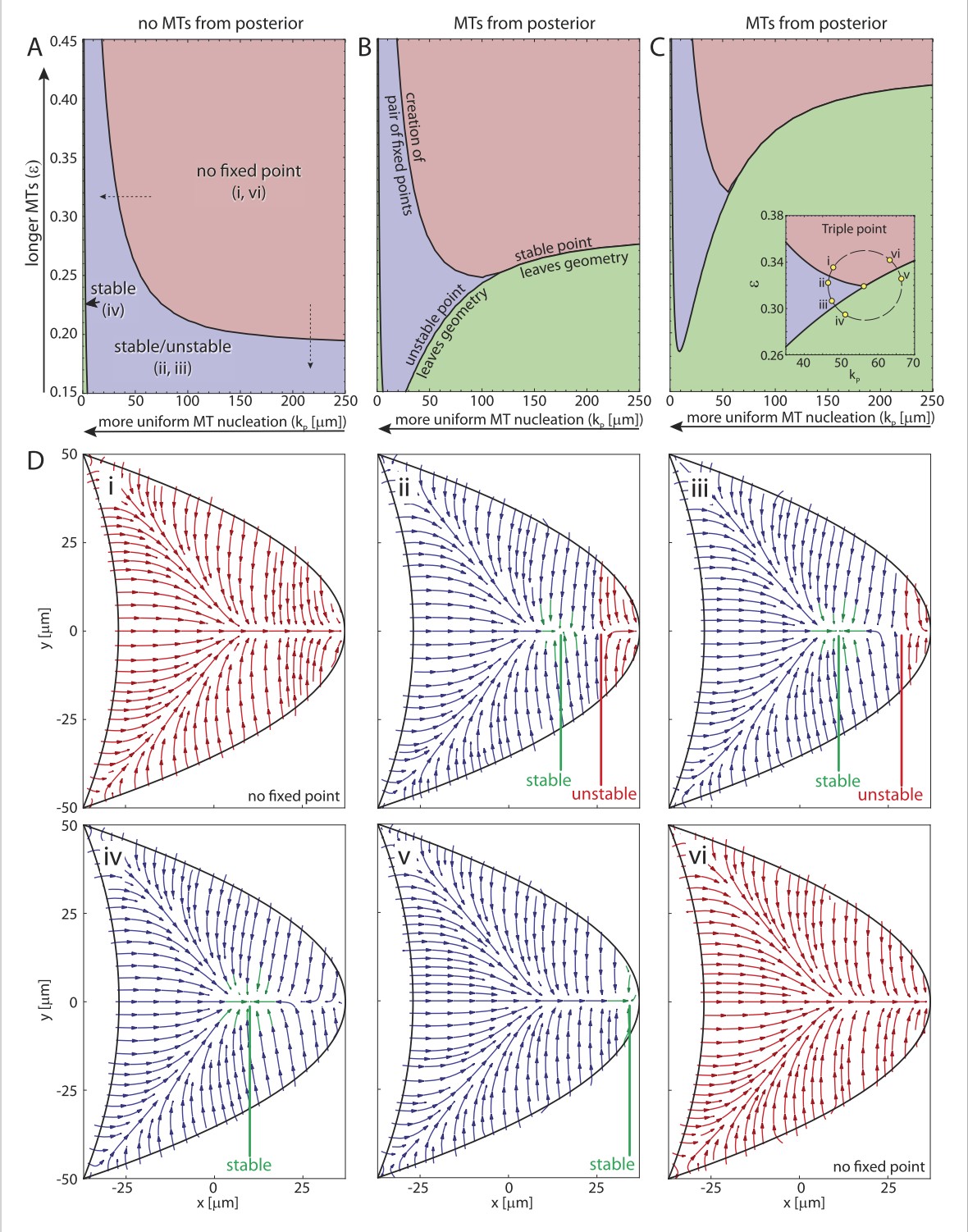

**Figure 5**. The parameter space of the rod model shows the relation and interconversion between all three distinct cytoskeletal architectures. (**A**) The regions corresponding to wild-type (red), weak *par-1* hypomorph (blue) and strong *par-1* hypomorph topologies (green, arrowhead at far left) are shown as a function of the mean MT length $\epsilon$ and extent of the posterior seeding density $k_P$. The bifurcation line between wild-type and weak *par-1* hypomorph topologies can be crossed by either changing the seeding density laterally (horizontal dashed arrow), or by shortening the MTs (vertical dashed arrow). (**B**, **C**) Parameter spaces as in **A** for increasing MT nucleation at the posterior pole (**B**: $h_0^P = 0.03$, **C**: $h_0^P = 0.1$). Inset in panel **C** shows a magnification of the triple point at which small changes in parameters can convert each cytoskeletal architecture into any other. (**D**) Local net orientations of MT rods for parameter values indicated in the inset of panel **C** (yellow circles). In all panels, parameter values for anterior MT nucleation were kept constant ($h_0^A = 0.8$, $k_A = 1000$ μm).

wild-type-like cytoskeletal topology that allows *oskar* mRNA localisation. Therefore, a low level of posterior nucleation of MTs does not necessarily lead to mislocalisation of *oskar* mRNA. Instead, and in addition to the distribution of cortical MT nucleation, the length of MT filaments emerges as another key regulator for polarisation and function of the non-centrosomal network.

## Discussion

Non-centrosomal MT networks represent a large, yet poorly understood class of MT arrangements that often fulfil specialised functions (*Keating and Borisy, 1999*). Non-centrosomal MTs are frequently aligned in parallel, thereby forming linear arrays as in epithelia or neurons (*Bartolini and Gundersen, 2006*; *Conde and Caceres, 2009*; *Kapitein and Hoogenraad, 2011*). The non-centrosomal MT cytoskeleton in *Drosophila* oocytes was also first hypothesised to form a highly polarised linear array with MTs growing from the anterior surface towards the posterior pole (*Clark et al., 1994*; *Brendza et al., 2000*). Instead, the MT cytoskeleton is a complex network without a visibly discernible polarity along the AP axis (*Theurkauf et al., 1992*; *Cha et al., 2001*, *2002*), and its organisation remained ambiguous (*MacDougall et al., 2003*; *Januschke et al., 2006*; *Zimyanin et al., 2008*; *Parton et al., 2011*).

We showed that in two different theoretical models, in which MTs grow from the oocyte cortex into the volume, the cytoskeleton features an on average compartmental organisation and is therefore more ordered than previously thought. In combination with cytoplasmic flows and diffusion, this cytoskeletal organisation successfully reproduces localisations of *oskar* and naive *bicoid* mRNAs in wild-type oocytes. Such a spatially varying, compartmental organisation suggests that a single statistical measure of polarity, derived by averaging data from the entire oocyte (*Zimyanin et al., 2008*; *Parton et al., 2011*), may not be a sufficient metric to characterise mRNA localisation.

Cytoplasmic flows may generally contribute to mRNA transport. Flows at later stages of oogenesis are fast and well-ordered, and occur concurrently with a late-phase enhancement of *oskar* mRNA accumulation (*Sinsimer et al., 2011*). This is consistent with a contribution of flows to mRNA localisation (*Glotzer et al., 1997*; *Forrest and Gavis, 2003*). At stage 9 of oocyte development, which is the focus of the work presented here, flows are slow and chaotic. At this stage, mutants with reduced flows showed *oskar* mRNA localising normally (*Serbus et al., 2005*), thus suggesting that slow and chaotic flows may not play the same role as in late oogenesis. However, flows in these mutants were merely reduced rather than abolished completely, and the measurements underestimated flow speeds (*Serbus et al., 2005*), hence leaving the interpretation of flows unclear. We here find that the effects of slow cytoplasmic flows on mRNA transport are negligible, and that cytoskeletal transport alone is sufficient for localisations of *oskar* and naive *bicoid* mRNAs. In this view, slow cytoplasmic flows arise primarily as inevitable physical byproduct of active motor-driven transport on the cytoskeleton rather than as an evolutionarily selected trait. This appears to mirror findings in the *Caenorhabditis elegans* zygote in which P-granules segregate by dissolution and condensation rather than via transport by cytoplasmic flows (*Brangwynne et al., 2009*).

Interestingly, neither MT-based transport alone nor combined with cytoplasmic flows and diffusion is sufficient to reproduce the anterior localisation of nurse cell conditioned *bicoid* mRNA irrespective of the position of injection into the oocyte (*Cha et al., 2001*). This Exuperantia-dependent mechanism may rely either on an unobserved population of MTs in the oocyte that can be specifically recognised by conditioned *bicoid* mRNA (*Cha et al., 2001*; *MacDougall et al., 2003*), or on an unknown MT-independent mechanism.

The central finding of our work is that gradients of cortical MT nucleation are sufficient for the assembly of a functional compartmentalised MT cytoskeleton in wild-type oocytes. While many non-centrosomal MT arrays are linear and emphasise questions about establishment and maintenance of parallel filament orientations (*Ehrhardt, 2008*; *Lindeboom et al., 2013*), our result stresses the need to understand gradients in MT nucleation as an alternative strategy for the assembly of functional non-centrosomal arrays. Whether MTs in *Drosophila* oocytes are differentially nucleated along the cortex itself or created elsewhere and then differentially anchored at the cortex remains an interesting open question, but both scenarios are compatible with our model. Understanding how this gradient is established will therefore depend on discovering how PAR-1 regulates MT interactions with the cortex.

Cortical MT gradients cannot only account for the wild-type cytoskeletal configuration but also for the phenotypes observed in a hierarchy of *par-1* hypomorphs. Modulation of MT gradients along the posterior-lateral cortex alone are sufficient to explain the splitting of *oskar* mRNA between the centre

and the posterior pole of the oocyte (*Doerflinger et al., 2006*) via a saddle-node bifurcation, suggesting that the anterior and posterior-lateral surfaces of the oocyte are functionally decoupled. Generally, bifurcations in temporal behaviour govern important qualitative transitions in many biological systems, such as the lactose network in *Escherichia coli* (*Ozbudak et al., 2004*), cell cycle in yeast (*Charvin et al., 2010*), and collapses of bacterial populations (*Dai et al., 2012*). In *Drosophila*, one important qualitative change is the temporal transition between stages 7/8 and 9. During this transition the MT cytoskeleton reorganises from uniform nucleation around the cortex and *oskar* mRNA in the centre to the AP MT nucleation gradient with *oskar* mRNA at the posterior. It is therefore tempting to speculate that the sequence of PAR-1 mutants and the underlying bifurcation represent static snapshots of this dynamic developmental transition in wild-type.

In conclusion, the present work provides a model that describes the assembly, organisation and function of the non-centrosomal MT array in *Drosophila* oocytes and directs future attention to the molecular mechanisms that enable differential MT nucleation or anchoring at the cortex.

## Materials and methods

### Coordinates and oocyte geometries

For calculation of the MT cytoskeleton, the AP axis is aligned with the *z*-axis of a cartesian coordinate system, and results are later shifted and rotated to align with the *x*-axis for visualization, for subsequent computations of cytoplasmic flows and simulations of cargo transport. Dimensional coordinates cover the ranges $x \in [-L, L]$, $y \in [-L, L]$ and $z \in [0, z_0 L]$ with length scale $L = 50$ μm. After nondimensionalization with scale $L$, coordinates are ranged as $x' \in [-1, 1]$, $y' \in [-1, 1]$ and $z' \in [0, z_0]$. Similarly, nondimensionalization of the shape parameter $k_P$ in the MT seeding density leads to $k_P' = k_P / L$, and all primes will be dropped subsequently. Methods figures are shown in nondimensional spatial coordinates, and reported parameter values are nondimensional unless noted otherwise.

The 3D geometry of a typical stage 9 *Drosophila* oocyte is defined as two parabolic, rotationally-symmetric caps (*Figure 1*). Anterior ($i = A$) and posterior ($i = P$) parabolic caps are parameterized in cylindrical coordinates $\rho = \sqrt{x^2 + y^2} \in [0, 1]$ and $\phi \in [0, 2\pi]$ as

$$\boldsymbol{\sigma}_i(\rho, \phi) = \rho \, \widehat{\mathbf{e}}_\rho + z_0^i \left(1 - \rho^2\right) \widehat{\mathbf{e}}_z \, , \tag{3}$$

where $\widehat{\mathbf{e}}_\rho = (\cos(\phi), \sin(\phi), 0)$ and $\widehat{\mathbf{e}}_z = (0, 0, 1)$ are the unit vectors in radial and *z*-direction, respectively. The line element along the parabola is calculated as

$$\mathrm{d}s_i = \left\| \frac{\partial \boldsymbol{\sigma}_i}{\partial \rho} \right\| \mathrm{d}\rho = \sqrt{1 + \left(2 z_0^i \, \rho\right)^2} \mathrm{d}\rho,$$

with arclength

$$s_i(\rho) = \frac{\rho}{2} \sqrt{1 + \left(2 z_0^i \, \rho\right)^2} + \frac{1}{4 z_0^i} \sinh^{-1}\left(2 z_0^i \, \rho\right), \tag{4}$$

defined such that $s_i(\rho = 0) = 0$ denotes the tip of the parabolic cap while $s_i(\rho = 1) = s_0^i$ denotes the distance from the tip to the anterior corners (*Figure 6A,G*), and hence $0 \leq s_i(\rho) \leq s_0^i$. The surface element for parabolic caps (in units of $L$) is given by

$$\mathrm{d}\Sigma_i = \left\| \frac{\partial \boldsymbol{\sigma}_i}{\partial \rho} \times \frac{\partial \boldsymbol{\sigma}_i}{\partial \phi} \right\| = \rho \, \sqrt{1 + \left(2 z_0^i \rho\right)^2} \mathrm{d}\rho \, \mathrm{d}\phi \, , \tag{5}$$

with total surface area

$$\Sigma_i = \frac{\pi}{6 \left(z_0^i\right)^2} \left( \left[1 + \left(2 z_0^i\right)^2\right]^{3/2} - 1 \right) \, . \tag{6}$$

The inward pointing normals for posterior and anterior caps are given by

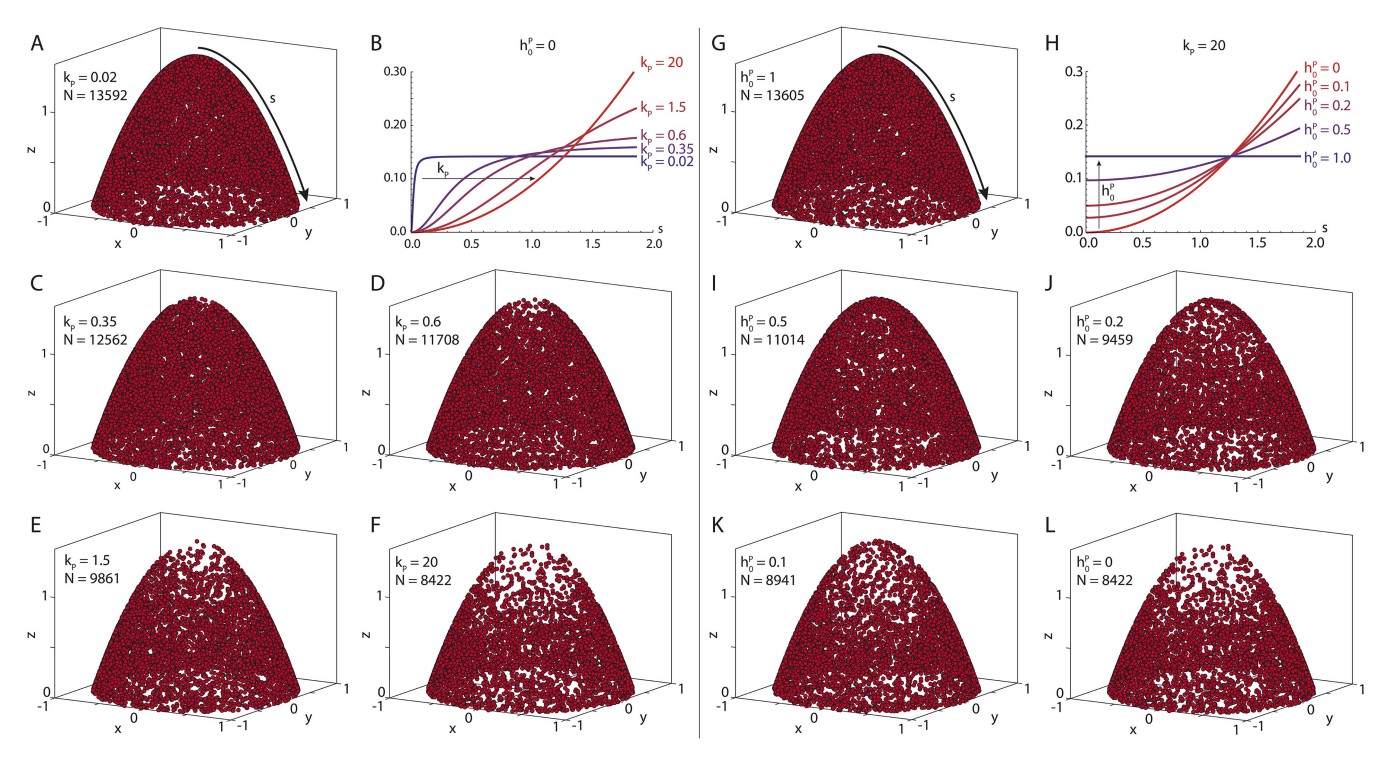

**Figure 6**. Seeding point density in the polymer model. Shown are $N$ randomly drawn seeding points on the posterior cap of the standard oocyte geometry, according to *Equation 15* distributed either (near-)uniformly on the cap (**A**, **G**) or in a parabolic gradient from the posterior pole to the anterior corners (**F**, **L**). Between uniform and parabolic distribution, the seeding density can vary either by laterally reducing the density (**B**, **C**–**E**, parameter $k_P$), or by reducing the density at the posterior pole (**H**, **I**–**K**, parameter $h_0^P$). Total number of points is calculated with a fixed number of anterior points $N_A = 4000$ (*Equation 20*). For actual computations of wild-type cytoskeletons ($k_P = 3$, $h_0^P = 0$, $k_A = 20$, $h_0^A = 0.8$, see main text for dimensional parameters), more than 55,000 seeding points are used.

$$\widehat{\boldsymbol{n}}_P(\rho, \phi) = -\frac{2z_0^P \, \rho \, \widehat{\boldsymbol{e}}_\rho + \widehat{\boldsymbol{e}}_z}{\sqrt{1 + \left(2z_0^P \rho\right)^2}} \ ,$$ (7)

$$\widehat{\boldsymbol{n}}_A(\rho, \phi) = \frac{2z_0^A \, \rho \, \widehat{\boldsymbol{e}}_\rho + \widehat{\boldsymbol{e}}_z}{\sqrt{1 + \left(2z_0^A \rho\right)^2}} \ .$$

A special case arises for $z_0^A = 0$ when the anterior parabolic cap becomes a flat disc. In this case, the arclength (*Equation 4*) reduces to $s_A(\rho) = \rho$, and the surface area (*Equation 6*) simplifies to $\Sigma_A = \pi$. The inwards pointing normals for this case are given by $\widehat{\boldsymbol{n}}_P$ (*Equation 7*) for the posterior cap and $\widehat{\boldsymbol{n}}_A = \widehat{\boldsymbol{e}}_z$ for the anterior disc, respectively.

To investigate the robustness with respect to changes in oocyte shape, we test our model for the cytoskeleton, cytoplasmic flows and mRNA transport in two different geometries for a stage 9 *Drosophila* oocyte. Geometry-1 is used as standard geometry, and is comprised of two parabolic caps (*Equation 3*) with $z_0^P = 1.48$ for the posterior cap and $z_0^A = 0.2$ for the anterior cap (see *Figure 1*). This results in a length of the AP axis of $z_0^P - z_0^A = 1.28$ with an aspect ratio of 1.56 that qualitatively resembles a typical stage 9 oocyte. Geometry-2 is tested as alternative geometry, and consists of a posterior parabolic cap with $z_0^P = 1$, and an anterior flat disc with $z_0^A = 0$ (*Figure 1—figure supplement 1*), giving an AP-axis length of 1 and aspect ratio of 2. We find that results are robust with respect to this change in geometry.

## Polymer model for MT cytoskeleton

### MT nucleation probability

The first step in the computation of the MT cytoskeleton is the generation of MT seeding points that are randomly positioned along the oocyte membrane. We define the following probability density along the arclength $s_i(\rho)$ of the anterior and posterior parabolic caps

$$p_\Sigma\left(s_i(\rho)\Big|h_0^i, k_i\right) = \frac{1}{A}\, \tilde{p}_\Sigma\left(s_i(\rho)\Big|h_0^i, k_i\right)\ , \tag{8}$$

with normalization over the oocyte surface

$$A = \int_{\Sigma_A} d\Sigma_A\, \tilde{p}_\Sigma\left(s_A(\rho)\Big|h_0^A, k_A\right) + \int_{\Sigma_P} d\Sigma_P\, \tilde{p}_\Sigma\left(s_P(\rho)\Big|h_0^P, k_P\right), \tag{9}$$

where the surface elements were specified in *Equation 5* and the arclength was given in *Equation 4*. For the functional form of $\tilde{p}_\Sigma(s(\rho)|h_0, k)$ we use the expression

$$\tilde{p}_\Sigma(s(\rho)|h_0, k) = h_0 + (1 - h_0)\left[1 + \left(\frac{k}{s_0}\right)^2\right]\frac{s(\rho)^2}{k^2 + s(\rho)^2}\ , \tag{10}$$

wherein the index $i$ was dropped for simplicity. Note that for the maximum arclength $s = s_0$ it holds that $\tilde{p}_\Sigma(s = s_0|h_0, k) = 1$.

$h_0 \in [0, 1]$ and $k \in (0, \infty)$ represent two parameters governing the shape of the probability density. For small values $k \ll s_0$ the nucleation density approaches a Hill function with Hill coefficient $n = 2$ and depth $h_0$

$$\lim_{k \ll s_0} \tilde{p}_\Sigma(s(\rho), |h_0, k) = h_0 + (1 - h_0)\frac{s(\rho)^2}{k^2 + s(\rho)^2}\ . \tag{11}$$

In the opposite limit $k \gg s_0$, the probability density approaches a parabola with depth $h_0$ (*Figure 6B,H*, red curve)

$$\lim_{k \gg s_0} \tilde{p}_\Sigma(s(\rho), |h_0, k) = h_0 + (1 - h_0)\left(\frac{s(\rho)}{s_0}\right)^2\ , \tag{12}$$

thus creating a pronounced gradient of MT nucleation from the pole ($s = 0$) to the corners ($s = s_0$).

MT nucleation along the arclength can increase from the gradient to a homogeneous nucleation along the arclength in two different ways: (i) either by increasing nucleation from the corners laterally towards the posterior pole, or (ii) by increasing nucleation uniformly at the posterior pole. Each shape parameter $h_0$ and $k$ accounts for one of these possibilities. Increasing the parameter $h_0$ leads to shallower gradients by increasing the probability for nucleation at the pole from 0 for $h_0 = 0$ to the value for homogeneous nucleation for $h_0 = 1$ (*Figure 6H*). Contrary, decreasing the parameter $k$ leads to narrower regions of low nucleation probability by decreasing its width laterally (*Figure 6B*).

To compare to experimental data, we stained $\alpha$-tubulin in fixed stage 9 wild-type *Drosophila* oocytes and extracted fluorescence intensity profiles along the cortex from the posterior pole to the anterior corners (*Figure 7A,B*). An ensemble of smoothed fluorescence profiles shows steep increases in cortical MT density from posterior to anterior, albeit with high variability. Still, a quadratic fit of the ensemble shows that a parabolic function is a plausible representation of the increasing MT density (*Figure 7C*). Several oocytes show fluorescent profiles that exhibit a local maximum at the posterior. This likely stems from out of plane fluorescence due to the geometric shape of the oocyte pole, and we neglect this minor effect here. At the anterior surface, MT density tends to be more homogeneous with only shallow increases in cortical MT density towards the corners. In summary, for the posterior cap we choose a probability density with parabolic shape that reaches zero at the posterior pole $h_0^P = 0, k_P = 20$ (*Figure 7D*, bottom), whereas for the anterior cap we chose a density with parabolic shape but only shallow depth $h_0^A = 0.8, k_P = 20$ (*Figure 7D*, top).

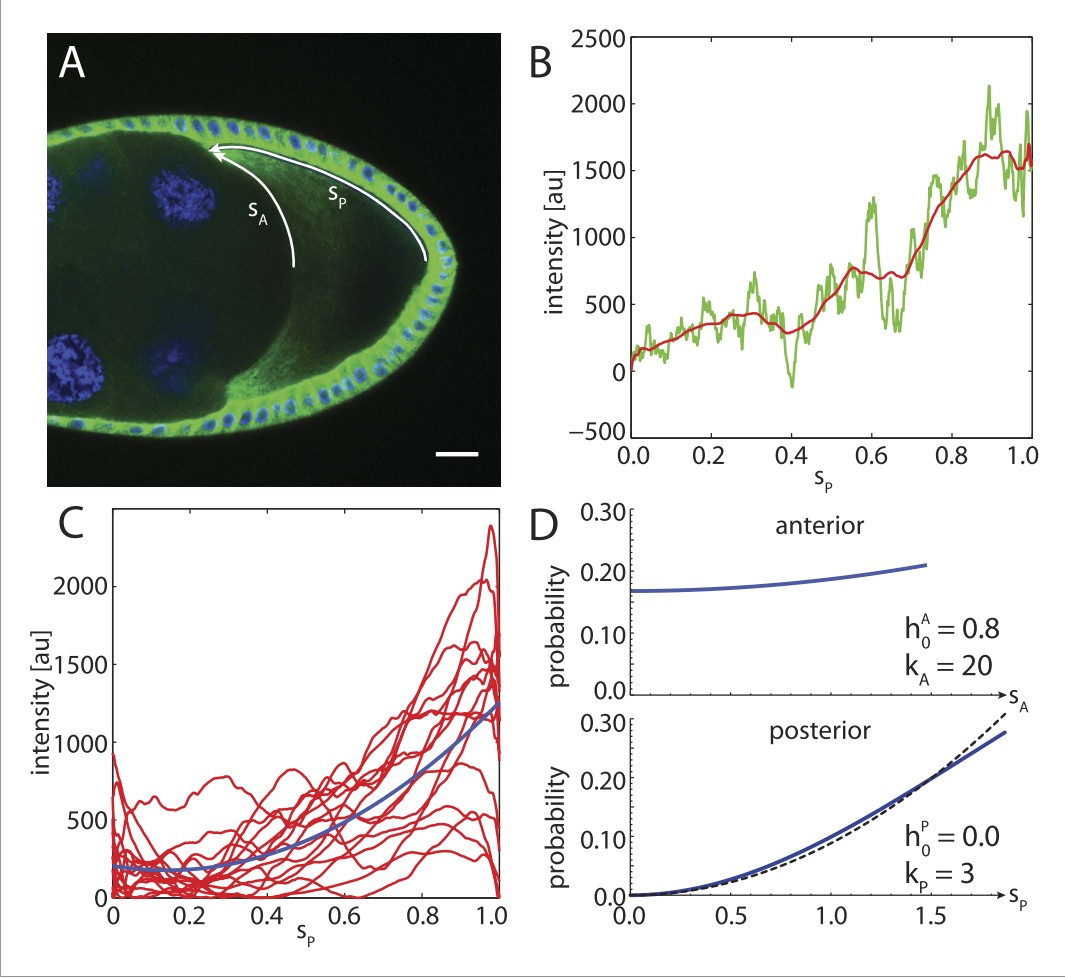

**Figure 7**. The cortical MT density increases approximately parabolically from the posterior pole towards the anterior corners. (**A**) $\alpha$-tubulin staining of a stage 9 oocyte (green) with DAPI staining of DNA (blue). Arrows indicate arclengths $s_P$ from the posterior pole to the anterior corners and analogously $s_A$ at the anterior. (**B**) Fluorescence intensity profile (green) with moving average (red) extracted from a 10 pixel wide line along $s_P$ indicated in panel **A**. Arclength was normalized to one, and the minimum intensity of the moving average was subtracted from both fluorescence profiles. (**C**) Average intensity profiles (red) of cortical MT density as in panel **B** from $N = 15$ oocytes. The minimum intensity value across all profiles was subtracted from each. Blue line shows a parabolic fit. (**D**) Probability densities for the distribution of MT seeding points along anterior arclength (top) and posterior arclength (bottom) used for the wild-type cytoskeleton (blue lines). Black dashed line in bottom panel shows exact parabola. Arclengths are nondimensional lengths with scale $L = 50$ μm.

## Generation of seeding points

Discrete seeding points that are randomly positioned along the oocyte membrane according to the probability density $p_\Sigma(s_i(\rho)|h_0^i, k_i)$ (**Equation 8**) can be achieved by inverse transform sampling. To this end, the cumulative distribution function on the anterior ($i = A$) or posterior ($i = P$) cap can be computed as

$$\mathscr{C}\left(\rho'\Big|h_0^i, k_i\right) = \int_0^{2\pi}\int_0^{\rho'} d\Sigma_i \, p_\Sigma\left(s_i(\rho)\Big|h_0^i, k_i\right) . \tag{13}$$

Note that $\mathscr{C}(\rho' = 1|h_0^i, k_i) \neq 1$ due to normalization across the total oocyte surface rather than across each individual cap. Therefore, random seeding points on each cap can be drawn by applying the inverted cumulative distribution function to a random sample that is uniformly distributed over the interval $[0, \mathscr{C}(\rho' = 1|h_0^i, k_i)]$. In this formulation, the total number of seeding

points is $N = N_A + N_P$ (see *Figure 6*). The ratio of anterior ($N_A$) and posterior ($N_P$) seeding points to be drawn must equal the ratio of total probability for a point to fall on the anterior and posterior caps, respectively. For a fixed value $N_A$, this results in

$$N_P = N_A \; \frac{\int d\Sigma_P \; p_\Sigma\left(s_P(\rho)\Big|h_0^P, k_P\right)}{\int d\Sigma_A \; p_\Sigma\left(s_A(\rho)\Big|h_0^A, k_A\right)} \, . \tag{14}$$

Here, we generated seeding points in an equivalent way by renormalizing the probability density (*Equation 8*) on each cap individually as

$$\mathscr{P}_\Sigma\left(s_i(\rho)\Big|h_0^i, k_i\right) = \frac{1}{\mathscr{A}} \; \tilde{p}_\Sigma\left(s_i(\rho)\Big|h_0^i, k_i\right) \, , \tag{15}$$

wherein the new normalization factor $\mathscr{A}$ is obtained by integration over only one parabolic cap

$$\mathscr{A}\left(h_0^i, k_i\right) = \int d\Sigma_i \; \tilde{p}_\Sigma\left(s_i(\rho)\Big|h_0^i, k_i\right) \, . \tag{16}$$

Hence, the corresponding cumulative distribution function

$$\mathscr{C}\left(\rho'\Big|h_0^i, k_i\right) = \int_0^{2\pi} \int_0^{\rho'} d\Sigma_i \; \mathscr{P}_\Sigma\left(s_i(\rho)\Big|h_0^i, k_i\right), \tag{17}$$

covers the range [0, 1] and seeding points are obtained as $\mathscr{C}^{-1}(u)$ for a random sample $u$ with uniform distribution $u \in [0, 1]$.

For parabolic caps, the normalization (*Equation 16*) can not be calculated analytically. Therefore, the probability density (*Equation 15*) and its cumulative distribution function (*Equation 17*) were evaluated and inverted numerically. For the special case of a flat anterior ($i = A$) disc in the alternative geometry-2, the normalization can be solved in closed form as

$$\mathscr{A}\left(h_0^A, k_A\right) = \int_0^{2\pi} d\phi \int_0^1 d\rho \; \rho \; \tilde{p}_\Sigma\left(\rho, h_0^A, k_A\right)$$

$$= \pi\left[\left(1 - h_0^A\right)k_A^2 + 1 - \left(1 - h_0^A\right)k_A^2\left(k_A^2 + 1\right)\ln\left(1 + 1\Big/k_A^2\right)\right] \, . \tag{18}$$

The corresponding cumulative distributions function evaluates to

$$\mathscr{C}(\rho') = \int_0^{2\pi} d\phi \int_0^{\rho'} d\rho \; \rho \; \mathscr{P}_\Sigma\left(\rho, h_0^A, k_A\right)$$

$$= \frac{\rho^2\left[\left(1 - h_0^A\right)k_A^2 + 1\right] - \left(1 - h_0^A\right)k_A^2\left(k_A^2 + 1\right)\ln\left[1 + (\rho/k_A)^2\right]}{\left(1 - h_0^A\right)k_A^2 + 1 - \left(1 - h_0^A\right)k_A^2\left(k_A^2 + 1\right)\ln\left[1 + (1/k_A)^2\right]} \, , \tag{19}$$

which was numerically inverted to generate randomly positioned seeding points on the anterior disc.

Due to the new individual normalization, the values of anterior and posterior probability densities differ at the contact line between anterior and posterior caps. Consider two rings with areas $dA_i$ along the contact line ($s = s_0^i$) on the anterior ($i = A$) and posterior ($i = P$) cap which contain $dN_i = N_i \, \mathscr{P}_\Sigma(s = s_0^i|h_0^i, k_i) \, dA_i$ seeding points. Setting the number of anterior points fixed, and enforcing the point densities to be equal in the corner rings $dN_A/dA_A = dN_P/dA_P$ for equal ring areas $dA_A = dA_P$ allows to compute the number of posterior seeding points as

$$N_P = N_A \frac{\mathscr{P}_\Sigma\left(s = s_0^A\Big|h_0^A, k_A\right)}{\mathscr{P}_\Sigma\left(s = s_0^P\Big|h_0^P, k_P\right)},$$

$$= N_A \frac{\mathscr{A}\left(h_0^P, k_P\right)}{\mathscr{A}\left(h_0^A, k_A\right)} ,\tag{20}$$

where the second line follows because $\tilde{p}_\Sigma(s_i = s_0^i | h_0^i, k_i) = 1$. Inserting the expression for $\mathscr{A}(h_0^i, k_i)$ (**Equation 16**) shows that forcing seeding point densities on the anterior and posterior caps to be equal at the anterior corners (**Equation 20**) is identical to chosing points according to the ratio of total probabilities on the anterior and posterior caps (**Equation 14**). Thus, chosing points from $p_\Sigma$ normalized over the entire surface (**Equation 8**) or from $\mathscr{P}_\Sigma$ normalized over each cap (**Equation 15**) is equivalent.

## Alternative seeding densities

Our model is not sensitive to the exact choice of the probability density for MT seeding points, and here we define alternative seeding densities that are mathematically tractable. Specifically, as randomly positioned seeding points are generated by operating the inverse CDF on a uniformly distributed sample $u \in [0, 1]$ and $\phi \in [0, 2\pi)$, it is convenient to chose the seeding probability such that the CDF can be inverted analytically.

We first note that writing out explicitly the CDF for a rotationally-symmetric probability density function $p(s_i(\rho))$ along the arclength (**Equation 13**) leads to

$$\mathscr{C}(\rho') = \int_0^{2\pi} d\phi \int_0^{\rho'} d\rho \; p(s_i(\rho)) \, \rho \, \sqrt{1 + \left(2z_0^i\rho\right)^2}.\tag{21}$$

For the case of a gradient of MT seeding points along the arclength of a parabolic cap we can define the density to be

$$p(s_i(\rho)) \equiv p_n\left(\rho | n, z_0^i\right) = \frac{(n+1)\, \rho^{n-1}}{2\pi \, \sqrt{1 + \left(2z_0^i\rho\right)^2}} .\tag{22}$$

From **Equation 21** this results in a CDF $u \equiv \mathscr{C}_n(\rho') = \rho'^{n+1}$ which can be analytically inverted as $\rho' = \mathscr{C}_n^{-1}(u) = u^{1/(n+1)}$.

For the case of a uniform distribution of seeding points along the arclength, the probability density is constant and given by the inverse surface area of the parabolic caps (**Equation 6**)

$$p(s_i(\rho)) \equiv p_u\left(\rho | z_0^i\right) = \frac{1}{\Sigma_i}.\tag{23}$$

For $p_u(\rho | z_0^i)$ we find the CDF

$$u \equiv \mathscr{C}_u(\rho') = \frac{\left(1 + \left(2z_0^i\rho'\right)^2\right)^{3/2} - 1}{\left(1 + \left(2z_0^i\right)^2\right)^{3/2} - 1} ,\tag{24}$$

which can again be inverted analytically to give

$$\mathscr{C}_u^{-1}(u) = \frac{1}{2z_0^i} \sqrt{\left(u\left[\left(1 + \left(2z_0^i\right)^2\right)^{3/2} - 1\right] + 1\right)^{2/3} - 1} .\tag{25}$$

For the special case of a flat disc, the uniform distribution of seeding points simplifies the CDF to

$$u \equiv \mathscr{C}_u = \int_0^{2\pi} d\phi \int_0^{\rho} d\rho \, \frac{1}{\pi} \, \rho = \rho^2,\tag{26}$$

with inverse $\rho = \mathscr{C}_u^{-1}(u) = \sqrt{u}$.

For a wild-type cytoskeleton, we impose a uniform MT density $p_u(\rho|z_0^i)$ along the anterior cap and a gradient of MTs $p_n(\rho|n, z_0^i)$ along the posterior cap. At the contact line between anterior and posterior caps, the seeding densities need to be matched by adjusting the total number of posterior seeding points $N_P$. A ring on the anterior cap at the anterior corners contains $dN_A = (N_A/\Sigma_A)dA_A$ seeding points, while a ring on the posterior cap at the anterior corners contains $dN_P = N_P\, p_n(\rho=1|n, z_0^P)\, dA_P$ points. Using *Equation 22* and *Equation 6* and forcing densities to be equal $dN_A/dA_A = dN_P/dA_P$ results in

$$N_P = \frac{N_A}{\Sigma_A\, p_n(\rho=1)}$$

$$= \frac{3(2z_0^A)^2 N_A}{n+1}\; \frac{\sqrt{1+(2z_0^P)^2}}{\left(1+(2z_0^A)^2\right)^{3/2} - 1}\; . \tag{27}$$

For the alternative geometry-2 comprised of a posterior parabolic cap and flat anterior disc, matching of seeding densities results in

$$N_P = \frac{2\,N_A}{n+1}\, \sqrt{1+(2z_0^P)^2}\; . \tag{28}$$

Using these alternative seeding densities does not qualitatively change the behavior of the model.

## MT growth

We describe MTs as persistent random walks, that is, a chain of straight segments of constant length $\lambda$ that show some flexibility in their relative orientations. The orientation $\hat{\boldsymbol{\mu}}_i$ of the $i$-th segment is drawn from a von Mises-Fisher probability distribution on a 2D sphere

$$f(\hat{\boldsymbol{\mu}}_i|\hat{\boldsymbol{\mu}}_{i-1}, \kappa) = \frac{\kappa}{4\pi\, \sinh(\kappa)}\; e^{\kappa\, \hat{\boldsymbol{\mu}}_i \cdot \hat{\boldsymbol{\mu}}_{i-1}}\; , \tag{29}$$

where $\kappa$ is the concentration parameter around the orientation $\hat{\boldsymbol{\mu}}_{i-1}$ of the previous segment (see 'Motor velocity field' for parameter values). The first orientation $\hat{\boldsymbol{\mu}}_1$ is drawn from the uniform angular distribution on a 2D sphere ($\kappa = 0$) where orientations are rejected if pointing locally outwards of the geometry $\hat{\boldsymbol{n}}_{A,P}(\rho, \phi) \cdot \hat{\boldsymbol{\mu}}_1 < 0$.

Polymers stop growing if they either encounter a boundary, or if they reach a pre-defined target length $l$ drawn from a probability distribution. The probability density is based on the experimental finding that MTs in vitro undergo a three-step aging process leading to catastrophe, resulting in catastrophe lengths ($l_c$) that follow a Gamma distribution (*Gardner et al., 2011*). From this catastrophe length distribution, the length distribution observed in an ensemble of growing MTs without shrinkage was calculated as

$$\phi_\Gamma(l|n, \Lambda) = \frac{\Gamma(n, l/\Lambda)}{n\, \Lambda\, \Gamma(n)}\; , \tag{30}$$

where $\Gamma(n, l)$ is the incomplete Gamma function, $\Gamma(n)$ is the Gamma distribution, $n = 3$ is the number of steps to reach catastrophe and $\Lambda$ is the step length (equivalent to a transition time per step for constant growth velocity, see *Gardner et al. (2011)*). We set the expectation value $E[\phi_\Gamma(l|n=3, \Lambda)] = 2\,\Lambda \equiv \mathcal{N}_\Gamma^{-1}$ as fraction $\epsilon$ of the AP axis length, hence $\Lambda = \epsilon(z_0^P - z_0^A)/2$.

Pseudo-random numbers distributed according to this probability density (*Equation 30*) were generated by inverse transform sampling of the cumulative distribution function

$$\Phi_\Gamma(l|n=3, \Lambda) = 1 - \frac{6\,\Lambda^2 + 4\,l\,\Lambda + l^2}{6\,\Lambda^2}\; e^{-l/\Lambda}\; . \tag{31}$$

Similar to the MT seeding densities, the model is not sensitive to the exact choice of the MT length distribution. For example, using the one-parameter exponential distribution

$$\phi_e(l|\Lambda) = \frac{1}{\Lambda} \, e^{-l/\Lambda}, \tag{32}$$

as an alternative length distribution and again regulating the expectation value $E[\phi_e(l|\Lambda)] = \Lambda = \epsilon(z_0^P - z_0^A)$ of MT lengths via a parameter $\epsilon$ does not qualitatively change our results.

## MT persistence length

The stiffness of a polymer is commonly characterized by its persistence length $P$, which is defined as the decay length of tangent–tangent correlations. To calculate the persistence length of MT random walks, we first define the end-to-end vector for a polymer with $N_s$ segments of length $\lambda$ and given initial orientation $\widehat{\boldsymbol{\mu}}_1$ as

$$\boldsymbol{R}(N_s|\widehat{\boldsymbol{\mu}}_1) = \lambda \sum_{i=1}^{N_s} \widehat{\boldsymbol{\mu}}_i. \tag{33}$$

Orientations are drawn from a von Mises-Fisher distribution (*Equation 29*) with mean

$$\langle \widehat{\mathbf{x}}|\widehat{\boldsymbol{\mu}}\rangle = \int \widehat{\mathbf{x}} \, f(\widehat{\mathbf{x}}|\widehat{\boldsymbol{\mu}},\kappa)d\widehat{\mathbf{x}} = \left(\frac{1}{\tanh(\kappa)} - \frac{1}{\kappa}\right)\widehat{\boldsymbol{\mu}} \equiv \sigma \, \widehat{\boldsymbol{\mu}} \, . \tag{34}$$

In order to calculate the persistence length of these polymers, we first consider the mean orientations of the second and third MT segments

$$\langle \widehat{\boldsymbol{\mu}}_2|\widehat{\boldsymbol{\mu}}_1\rangle = \int \widehat{\boldsymbol{\mu}}_2 \, f(\widehat{\boldsymbol{\mu}}_2|\widehat{\boldsymbol{\mu}}_1,\kappa)d\widehat{\boldsymbol{\mu}}_2 = \sigma \, \widehat{\boldsymbol{\mu}}_1 \, ,$$

$$\langle \widehat{\boldsymbol{\mu}}_3|\widehat{\boldsymbol{\mu}}_1\rangle = \int \widehat{\boldsymbol{\mu}}_3 \prod_{i=2}^{3} f(\widehat{\boldsymbol{\mu}}_i|\widehat{\boldsymbol{\mu}}_{i-1},\kappa)d\widehat{\boldsymbol{\mu}}_i = \sigma^2 \, \widehat{\boldsymbol{\mu}}_1 \, ,$$

and we find in general the relation

$$\langle \widehat{\boldsymbol{\mu}}_{N_s}|\widehat{\boldsymbol{\mu}}_1\rangle = \int \widehat{\boldsymbol{\mu}}_{N_s} \prod_{i=2}^{N_s} f(\widehat{\boldsymbol{\mu}}_i|\widehat{\boldsymbol{\mu}}_{i-1},\kappa)d\widehat{\boldsymbol{\mu}}_i = \sigma^{N_s-1} \, \widehat{\boldsymbol{\mu}}_1 \, . \tag{35}$$

Hence, using *Equation 35* and shifting indices, the expectation value of the end-to-end vector $\langle \boldsymbol{R}(N_s|\widehat{\boldsymbol{\mu}}_1)\rangle$ is given by

$$\lambda \sum_{i=1}^{N_s} \langle \widehat{\boldsymbol{\mu}}_i|\widehat{\boldsymbol{\mu}}_1\rangle = \lambda \sum_{n=0}^{N_s-1} \sigma^n \widehat{\boldsymbol{\mu}}_1 = \lambda \frac{1-\sigma^{N_s}}{1-\sigma} \, \widehat{\boldsymbol{\mu}}_1 \, . \tag{36}$$

For stiffness parameter $\kappa \to 0$ ($\sigma \to 0$), the von Mises-Fisher distribution becomes a uniform distribution on a 2D sphere. In this case, the polymers curl up with mean end-to-end distance of a single segment

$$\lim_{\kappa\to0}\langle \boldsymbol{R}(N_s|\widehat{\boldsymbol{\mu}}_1)\rangle = \lim_{\sigma\to0}\langle \boldsymbol{R}(N_s|\widehat{\boldsymbol{\mu}}_1)\rangle = \lambda \, \widehat{\boldsymbol{\mu}}_1 \, . \tag{37}$$

In the opposite limit $\kappa \to \infty$ ($\sigma \to 1$), the polymers become stiff rods and are extended to their maximum possible length

$$\lim_{\kappa\to\infty}\langle \boldsymbol{R}(N_s|\widehat{\boldsymbol{\mu}}_1)\rangle = \lim_{\sigma\to1}\langle \boldsymbol{R}(N_s|\widehat{\boldsymbol{\mu}}_1)\rangle = \lambda \, N_s \, \widehat{\boldsymbol{\mu}}_1 \, , \tag{38}$$

thereby showing that the polymers interpolate between an undirected random walk and completely straight rods.

The persistence length $P$ is defined as the spatial decay length of the correlations of polymer segment orientations $e^{-L/P} = \langle \cos(\theta)\rangle \equiv \langle \widehat{\boldsymbol{\mu}}_{N_s}\cdot\widehat{\boldsymbol{\mu}}_1\rangle$. For large $L = N_s\lambda$, we have

$$\frac{1}{P} = -\lim_{L\to\infty}\frac{1}{L} \, \ln\left(\langle \widehat{\boldsymbol{\mu}}_{N_s}\cdot\widehat{\boldsymbol{\mu}}_1\rangle\right)$$

$$= -\lim_{N_s \to \infty} \frac{1}{\lambda\, N_s} \ln\left(\sigma^{N_s-1}\right)$$

$$\approx -\frac{1}{\lambda} \ln(\sigma) \; . \tag{39}$$

For $\kappa \ll 1$, it holds that $\sigma \approx \kappa/3 + \mathcal{O}(\kappa^3)$ and hence

$$P \approx \frac{\lambda}{\ln(3/\kappa)} \; , \tag{40}$$

whereas for $\kappa \gg 1$ we find

$$P \approx \lambda\, \kappa \; . \tag{41}$$

Therefore, the persistence length is approximated as segment length $\lambda$ multiplied by concentration parameter $\kappa$.

## Motor velocity field

From each realization of the MT cytoskeleton in the polymer model, we derive a nondimensional vector field termed motor-velocity field $\boldsymbol{v}'_m$ by computing the local vectorial sum of MT segments. To this end, we define a coarse-grained cubic grid with nondimensional side length $dG = 0.04$ ranging from $-1 + dG/2$ to $1 - dG/2$ in x- and y-direction, and from $z_0^A + dG/2$ to $z_0^P - dG/2$ in z-direction. The center point for each MT segment is computed, and the orientations for all MT segments whose center points fall within the same grid volume are vectorially summed to give the average local MT orientation. By normalizing the resulting vector field to a mean vector magnitude of 1, we finally construct the motor-velocity field $\boldsymbol{v}'_m$. $\boldsymbol{v}'_m$ is used subsequently to compute the cytoplasmic flow field, and to simulate active motor-driven cargo transport on the cytoskeleton.

## Parameter values

For all computations of the MT cytoskeleton, we used $N_A$ = 25000 anterior seeding points, giving a total number of points between $N$ = 55764 for wild-type cytoskeleton and $N$ = 84953 for *par-1* null mutants.

The length of an individual MT segment $\lambda$ needs to be short compared to the system length $L$, yet long enough to allow simulations of even long MTs with feasible computational effort. Here, we chose $\lambda = 0.015$ μm. The maximum number of segments per MT is set to $N_s^{max}$ = 200. Care was taken that few MTs reach the maximum length $\lambda\, N_s^{max}$.

Experimental measurements of MT persistence lengths $P$ are on the order of millimeters (*van Mameren et al., 2009*). However, the effective persistence lengths of MTs in oocytes is likely shorter due to effects of cytoplasmic flows as well as high density of yolk granules and other obstacles in the cytoplasm that induce MT bending. MTs in the oocyte are neither seen to be completely straight (*Figure 8*, right), nor strongly curled up (*Figure 8*, left). Therefore, we chose an intermediate value of $\kappa$ = 18, corresponding to an effective persistence length (*Equation 41*) of $P \approx 13.5$ μm (*Figure 8*, center).

Due to their curved nature, the reach of a MT polymers is effectively shorter than the length of a straight rod composed of the same number of segments. For example, for the chosen values of $\kappa$ and $\lambda$, the end-to-end distance of a MT polymer with $N$ = 25 segments (*Equation 36*) on average 54% as long as a fully extended polymer or straight rod. Therefore, to compare the MT polymer model to a model in which MTs are represented as straight rods necessitates an adjustment in the mean MT length.

## Rod model for the MT cytoskeleton

### Model setup

In addition to the polymer model for MTs, we test the net orientation of the cytoskeleton by comparing with a second model in which MTs are nucleated continuously around the cell membrane and act as straight rods with a given length distribution. A single MT from a point $\sigma(\rho, \phi)$ on the boundary that hits an observation point $\boldsymbol{x}$ in the volume contributes an orientation vector pointing

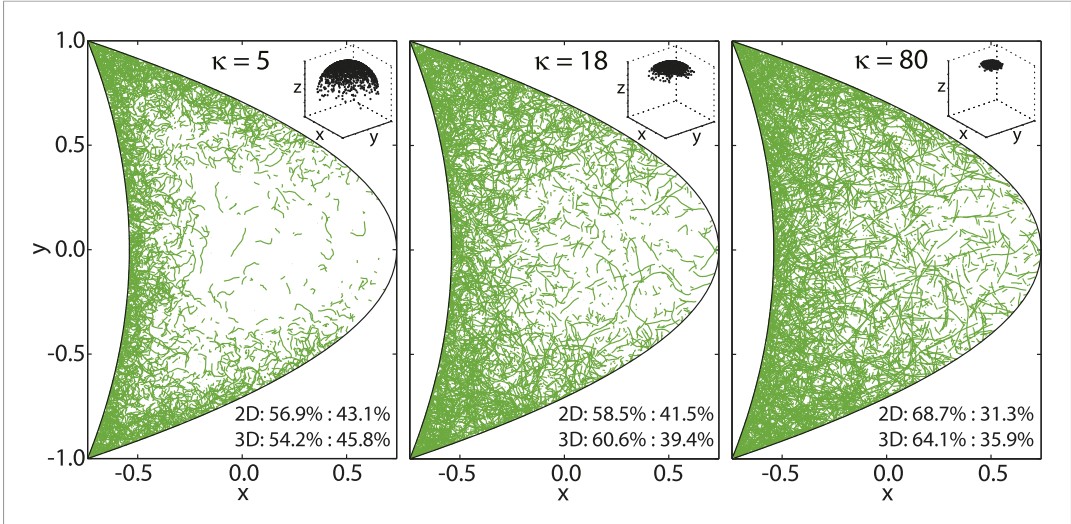

**Figure 8**. von Mises-Fisher parameter $\kappa$ determines MT stiffness. Shown are three cross sections through MT cytoskeletons for indicated values of $\kappa$. MTs become stiffer with increasing values of $\kappa$. Insets show angular distribution for $3 \times 10^3$ points drawn from von Mises-Fisher distribution around mean direction $\widehat{\boldsymbol{\mu}} = \widehat{\boldsymbol{e}}_z$.

from $\boldsymbol{\sigma}$ to $\boldsymbol{x}$ to the observation point. Heuristically, this contribution is expected to be weighted by three different factors: (i) the probability density of MT nucleation $p_\Sigma(\boldsymbol{\sigma})$ at boundary point $\boldsymbol{\sigma}(\rho, \phi)$, (ii) the probability that the nucleated MT rod is oriented in a direction such that it hits the observation point $\boldsymbol{x}$, and (iii) the probability that the MT rod is at least long enough to reach from $\boldsymbol{\sigma}$ to $\boldsymbol{x}$ across a distance $r_\sigma = \|\boldsymbol{x} - \boldsymbol{\sigma}\|$. Summing up all the weighted orientational contributions from the entire boundary eventually gives the net orientation at a given observation point inside the oocyte.

We denote the joint probability that a MT rod from the surface element $d\Sigma$ around a point $\boldsymbol{\sigma}$ reaches a volume element $dV$ around $\boldsymbol{x}$ by $p(\boldsymbol{x}, \boldsymbol{\sigma})d\Sigma\, dV$. $p(\boldsymbol{x}, \boldsymbol{\sigma})$ is normalized over the entire oocyte surface area $\Sigma$ and volume $V$ as

$$\int_\Sigma d\Sigma \int_V dV\, p(\boldsymbol{x}, \boldsymbol{\sigma}) = 1\ . \tag{42}$$

The marginal distribution $p_\Sigma(\boldsymbol{\sigma})$ given by

$$p_\Sigma(\boldsymbol{\sigma}) = \int_V dV\, p(\boldsymbol{x}, \boldsymbol{\sigma}), \tag{43}$$

defines the probability density of MT nucleation on the oocyte surface. Conversely, the marginal distribution $p_V(\boldsymbol{x})$

$$p_V(\boldsymbol{x}) = \int_\Sigma d\Sigma\, p(\boldsymbol{x}, \boldsymbol{\sigma}), \tag{44}$$

is proportional to the total density of MTs that reach the observation point $\boldsymbol{x}$ from the entire boundary. Each individual MT rod contributes an orientation unit vector $\widehat{\boldsymbol{e}}_{x\sigma} = (\boldsymbol{x} - \boldsymbol{\sigma})/\|\boldsymbol{x} - \boldsymbol{\sigma}\|$. Thus, the net MT orientation at observation point $\boldsymbol{x}$ corrected for the density of contributing MTs can be calculated as

$$\boldsymbol{o}(\boldsymbol{x}) = \frac{1}{p_V(\boldsymbol{x})} \int_\Sigma d\Sigma\, \widehat{\boldsymbol{e}}_{x\sigma}\, p(\boldsymbol{x}, \boldsymbol{\sigma})\ . \tag{45}$$

It is convenient to split the joint probability $p(\boldsymbol{x}, \boldsymbol{\sigma})$ into the conditional probability and the probability for MT nucleation at the surface

$$p(\boldsymbol{x}, \boldsymbol{\sigma}) = p(\boldsymbol{x}|\boldsymbol{\sigma})\, p_\Sigma(\boldsymbol{\sigma}). \tag{46}$$

For the nucleation of MTs on the oocyte surface $p_\Sigma(\boldsymbol{\sigma})$, we use the same probability density as in the polymer model (*Equation 8*). To specify the remaining conditional probability $p(\boldsymbol{x}|\boldsymbol{\sigma})$, first note that inserting *Equation 46* into *Equation 43* imposes the normalization condition

$$1 = \int_V dV\, p(\boldsymbol{x}|\boldsymbol{\sigma}). \tag{47}$$

Consider a local spherical polar coordinate system $(r_\sigma, \theta_\sigma, \phi_\sigma)$ centered at a given point $\boldsymbol{\sigma}$ on the oocyte surface with the $z_\sigma$-axis identical to the locally inwards pointing normal $\hat{n}_\sigma$. MT rods are assumed to be oriented in any direction into the positive half space $z_\sigma > 0$ with equal probability. To determine the general structure of the conditional probability density function $p(\boldsymbol{x}|\boldsymbol{\sigma})$, we first assume MT rods of cross sectional area $A$ and fixed length $l$ corresponding to a length distribution

$$\phi_\delta(r_\sigma) = \delta(l - r_\sigma), \tag{48}$$

with cumulative distribution function

$$\Phi_\delta(r_\sigma) = \int_0^{r_\sigma} dr\, \delta(l - r) = \Theta(r_\sigma - l)$$

$$= 1 - \Theta(l - r_\sigma). \tag{49}$$

The volume $V_c = A\, l$ of such a cylindrical MT rod can be approximated as integral over the positive half space by

$$V_c = A \int_0^l dr_\sigma = A \int_0^\infty dr_\sigma\, \Theta(l - r_\sigma)$$

$$= A \int_0^{2\pi} d\phi_\sigma \int_0^{\pi/2} d\theta_\sigma\, \sin(\theta_\sigma) \int_0^\infty dr_\sigma\, r_\sigma^2 \frac{\Theta(l - r_\sigma)}{2\pi r_\sigma^2},$$

$$\Leftrightarrow 1 = \int_V dV\, \frac{\Theta(l - r_\sigma)}{2\pi r_\sigma^2 l}\,, \tag{50}$$

where the last step resulted from division by $A\, l$. Comparing this result with *Equation 47* and using *Equation 49*, we identify the conditional probability density function for randomly oriented MTs rods of fixed length as

$$p(\boldsymbol{x}|\boldsymbol{\sigma}) = \mathscr{N}_\delta\, \frac{1}{2\pi r_\sigma^2} [1 - \Phi_\delta(r_\sigma)]\,. \tag{51}$$

Herein, the factor $\mathscr{N}_\delta = 1/l$ corresponds to the mean MT length, the term $(2\pi r_\sigma^2)^{-1}$ represents the uniform angular distribution on a half sphere, and the expression $[1 - \Phi(r_\sigma)]$ is the probability that MT rods are at least $r_\sigma$ long.

To account for the experimentally measured MT length distribution, *Equation 51* is finally generalized by replacing $\Phi_\delta(r_\sigma)$ by the cumulative distribution used in the polymer model $\Phi_\Gamma(r_\sigma|n = 3, \Lambda)$ (*Equation 31*) with $\mathscr{N}_\Gamma = 1/(2\Lambda)$. Thus, the complete joint probability distribution $p(\boldsymbol{x}, \boldsymbol{\sigma})$ (*Equation 46*) for a MT emanating from $\boldsymbol{\sigma}(\rho)$ and contributing to the net orientation at observation point $\boldsymbol{x}$ in the oocyte volume is defined as

$$p(\boldsymbol{x}, \boldsymbol{\sigma}) = p_\Sigma(s|h_0, k)\, \mathscr{N}_\Gamma\, \frac{1}{2\pi r_\sigma^2} [1 - \Phi_\Gamma(r_\sigma|3, \Lambda)]\,. \tag{52}$$

Note that MT straight rods always cover the distance between nucleation and observation point in a straight line. If the anterior surface is curved inwards, these straight lines can temporarily pass outside of the oocyte geometry before reaching the observation point inside the volume again.

The rod model does not prevent such unphysical contributions (though the polymer model does prevent this). However, this error does not occur in a geometry with a flat anterior boundary. Calculation of the cytoskeleton in a geometry with flat anterior (*Figure 1—figure supplement 1G*) does not change the observed topology, suggesting that in practice this error is negligible.

## Parameter space and comparison to polymer model

The continuum description of the rod model allows one to follow the saddle-node bifurcation in the MT cytoskeleton precisely throughout parameter space. The creation of stable and unstable fixed points always occurs along the AP axis because the system is rotationally symmetric. We therefore compute the uncorrected net orientation $o(x)p_V(x)$ (*Equation 45*) for observation points $x$ only along the symmetry axis as a function of cortical nucleation parameter $k_P$ and mean MT length $\epsilon$.

For a cytoskeletal topology without any fixed point, the x-component $o_x(x)p_V(x)$ will be pointing towards the posterior pole everywhere along the AP axis. A positive value $o_x(x)p_V(x) > 0$ everwhere along the central axis therefore defines the regions of a wild-type cytoskeletal topology. If a pair of a stable and a saddle-node is present, however, $o_x(x)p_V(x)$ will be negative at least one point along the AP axis, while at the same time it will be positive at the grid point closest to the posterior pole $x_P$ where the unstable node creates a limited domain of attraction. Combination of both criteria therefore defines the region of parameter space in which the oocyte is split into two different regions of attraction. Finally, a negative value at the posterior pole $o_x(x_P)p_V(x_P) < 0$ is the signature of a cytoskeleton with only a stable point whose domain of attraction covers the entire volume (see *Figure 5*).

In the ensemble averages of the polymer model, we find that the bifurcation point occurs for small values of $k_P$ if $\epsilon$ is large, but shifts to larger values of $k_P$ if $\epsilon$ becomes smaller. Furthermore, the bifurcation point can be crossed solely by varying $\epsilon$ while keeping $k_P$ fixed (*Figure 4—figure supplement 1*). In summary, this confirms that the polymer model exhibits the same parameter space structure as the rod model (*Figure 5*), thus further supporting the similarity of the two models. Moreover, as in the rod model, this parameter space structure is again insensitive to the oocyte shape.

## Bifurcation normal form

The saddle-node bifurcation normal form in two dimensions is given by

$$\begin{pmatrix} v_x \\ v_y \end{pmatrix} = \begin{pmatrix} x^2 + \lambda \\ -y \end{pmatrix} , \tag{53}$$

wherein $\lambda$ is the bifurcation parameter. For negative $\lambda$, stable and unstable fixed points exist and are located at $x_\pm = \pm\sqrt{-\lambda}$. The critical bifurcation point occurs at $\lambda = 0$ (*Figure 4—figure supplement 1P–T*).

## Additional methods

### Nondimensionalization and mathematical methods

In dimensional units, the system of transport equations for bound and unbound cargo fractions $c_b$ and $c_u$ is defined as

$$\partial_t\, c_b + \nabla \cdot (\boldsymbol{v}_m\, c_b) = k_b\, c_u - k_u\, c_b, \tag{54}$$

$$\partial_t\, c_u + \nabla \cdot (\boldsymbol{u}\, c_u) = -k_b\, c_u + k_u\, c_b + D\, \nabla^2 c_u , $$

while the Navier-Stokes-Equations and volume forces are defined by

$$\rho(\partial_t\, \boldsymbol{u} + (\boldsymbol{u} \cdot \nabla)\boldsymbol{u}) = -\nabla p + \mu\, \nabla^2 \boldsymbol{u} + \boldsymbol{f}, \tag{55}$$

$$\boldsymbol{f} = a\, \boldsymbol{v}_m , \tag{56}$$

with no-slip boundary conditions on the oocyte surface. Note that cytoplasmic streaming is present in oocytes even when *oskar* mRNA is not expressed (*Palacios and St Johnston, 2002*), showing that some cargo other than *oskar* mRNA is transported by kinesin and responsible for driving flow. This justifies that the parameter $a$ in *Equation 56* is independent of the concentration of the bound cargo $c_b$.

Computations of the fluid flow field and simulations of cargo transport are performed using a nondimensionalized version of *Equations 54–56*. For nondimensionalization, we select a length scale $L = 50\ \mu\text{m}$ for scaling of space $\boldsymbol{x} = L\,\boldsymbol{x}'$, and an advection scale derived from the active motor-driven

transport $V = 0.5$ μm/s (**Zimyanin et al., 2008**) for scaling of velocities $\boldsymbol{v}_m = V \boldsymbol{v}_m'$, implying an advection time scale $\tau = L/V = 100$ s for scaling of time $t = \tau t'$. $\boldsymbol{v}_m'$ denotes the coarse-grained motor-velocity field computed from each realization of the MT polymer model ('MT persistence length'). Defining the mean raction rate constant $K = (k_b + k_u)/2$ and a nondimensional reaction parameter $\beta = k_b/(2 K)$ we find the nondimensional version of **Equation 54**

$$\partial_{t'} c_b' + \nabla' \cdot \left( \boldsymbol{v}_m'\, c_b' \right) = 2\mathrm{Da}\left( \beta c_u' - (1 - \beta) c_b' \right), \tag{57}$$

$$\partial_{t'} c_u' + \nabla' \cdot \left( \boldsymbol{u}'\, c_u' \right) = 2\mathrm{Da}\left( -\beta c_u' + (1 - \beta) c_b' \right) + \mathrm{Pe}^{-1} \nabla'^2 c_u' \,,$$

where $\mathrm{Da} = L K/V$ is the Damköhler number and $\mathrm{Pe} = L V/D$ is the Peclet number.

For nondimensionalization of **Equations 55, 56**, we define scales for the pressure $\mathbb{P} = \mu V/L$, force density $\mathbb{F} = \mu V/L^2$ and force-velocity scaling $\mathbb{A} = \mathbb{F}/V$ to obtain

$$\mathrm{Re}\left( \partial_t'\, \boldsymbol{u}' + (\boldsymbol{u}' \cdot \nabla') \boldsymbol{u}' \right) = -\nabla' p' + \nabla'^2 \boldsymbol{u}' + \boldsymbol{f}', \tag{58}$$

$$\boldsymbol{f}' = a'\, \boldsymbol{v}_m', \tag{59}$$

wherein $p' = p/\mathbb{P}$, $\boldsymbol{f}' = \boldsymbol{f}/\mathbb{F}$, $a' = a/\mathbb{A}$, and $\mathrm{Re} = \rho V L/\mu$ is the Reynolds number. Note that the Reynolds number is defined using the scale $V$ of the active transport field because the flow velocity field $\boldsymbol{u}$ is derived from it and hence varies. To estimate an upper bound on the Reynolds number, we use the viscosity of water $\mu = 10^{-3}$ Pa s and 10 times the density of water $\rho = 10^4$ kg/m$^3$ as lower and upper bounds for the viscosity and density of the cytoplasm in the oocyte. This results in an upper bound $\mathrm{Re} = 2.5 \times 10^{-4}$, thus justifying neglecting the inertia terms in **Equation 58**.

## Parameter values

The unbinding constant $k_u$ can be estimated as ratio of the mean active transport velocity and the mean track length of *oskar* mRNA in stage 9 oocytes as $k_u = 0.17$ s$^{-1}$ (**Zimyanin et al., 2008**). The nondimensional parameter $\beta$ determines the fraction of cargo that resides in the bound state. Given the fact that 13% of *oskar* mRNA are bound at any given time we set $\beta = 0.13$ (**Zimyanin et al., 2008**). The unbinding constant $k_u$ and $\beta$ can be used to calculate $k_b = 0.0255$ s$^{-1}$. Therefore, the mean reaction rate constant and the Damkoehler number are $K = 0.0978$ and $\mathrm{Da} = 9.775$, respectively. Diffusion of mRNP particles in the ooplasm is widely considered to be very slow. Experimentally measured diffusion constants for mRNA transcripts of sizes similar to *oskar* mRNA (2.9 kB) in mammalian cells are in the range of $D = 0.02$ μm/s or lower (**Mor et al., 2010**). With this value, the Peclet number evaluates to $\mathrm{Pe} = 1250$.

The nondimensional scaling parameter $a'$ (**Equation 56**) which converts the active transport velocities into forces acting on the cytoplasmic fluid absorbs several unknown quantities including the size of the cargo that drives fluid flow, drag forces on the fluid with local viscosity, the density of such cargo–motor complexes on MTs and any collective effects. The value of $a'$ determines the mean nondimensional and dimensional fluid flow speeds $\langle |\boldsymbol{u}'| \rangle$ and $V \langle |\boldsymbol{u}'| \rangle$. We regard $a'$ as a macroscopic, phenomenological parameter. The value $a' = 45$ ($a' = 55$ for alternative geometry-2) is calibrated such that the resulting mean dimensional fluid flow speeds $V \langle |\boldsymbol{u}'| \rangle$ match the average fluid flow speeds measured experimentally for stage 9 *Drosophila* oocytes (see **Figure 2**) (compare to similar approach in **He et al. (2014)**).

To test the anchoring mechanism, the bound state is replaced by an anchored state to which cargo can only bind at the extreme posterior pole. For the anchoring state we set $k_u^{\mathrm{anch}} = 0$ s$^{-1}$ and $k_b^{\mathrm{anch}} = 10\, k_b$, resulting in $\beta^{\mathrm{anch}} = 1$ (no unbinding), $K^{\mathrm{anch}} = 0.1275$, and $\mathrm{Da}^{\mathrm{anch}} = 12.75$.

## Computational methods

For computation of the cytoplasmic flow field and subsequent transport simulations, the motor-velocity field $\boldsymbol{v}_m'$ and forces $\boldsymbol{f}'$ from the MT polymer model are shifted and rotated such that the AP axis aligns with the x-axis and the origin of the coordinate system is located in the oocyte center. To compute the cytoplasmic flow field we use the open source FreeFem++ finite element solver (**Hecht, 2012**). Forces $\boldsymbol{f}$ defined on a 3D cubic grid are interpolated to the unstructured 3D grid used by FreeFem++, and resulting flow velocity vectors are interpolated back to the cubic grid and derived staggered grids.

Cargo simulations of *Equations 57* are performed using custom written Matlab code in finite volume formulation on staggered grids as described in *Versteeg and Malalasekera (2007)* (custom Matlab code available at http://www.damtp.cam.ac.uk/user/gold/datarequests.html). As nondimensional time step we used $\Delta t = 0.005$. Each simulation cycles through many pairs of active transport fields $\boldsymbol{v}'_m$ and their corresponding cytoplasmic flow fields $\boldsymbol{u}'$ to account for a dynamically remodelling cytoskeleton and temporally changing cytoplasmic flow fields. Each pair of $\boldsymbol{v}'_m$-$\boldsymbol{u}'$-fields is active for 432 time steps, corresponding to 3.6 min of simulated real time. The set of $\boldsymbol{v}'_m$-$\boldsymbol{u}'$-fields is pre-computed before the start of the transport simulation.

## Fly stocks and experimental methods

Oocyte MTs were stained as described by *Theurkauf et al. (1992)* using a FITC-coupled anti-alpha-tubulin antibody at 1:200 (Sigma-Aldrich, St. Louis, MO). The general shape and size of the oocyte was imaged by recording the fluorescence from a *par-1* protein trap (PT) GFP line (*Lighthouse et al., 2008*). Homozygous or heterozygous *par-1* PT flies were also used for measurements of cytoplasmic flows by imaging movements of autofluorescent yolk granules. The allelic combination *par-1^W3*/*par-1^6323* was used for imaging cytoplasmic flows in strong *par-1* hypomorphs (*Shulman et al., 2000*). In all cases of flow measurements, flies were yeast fed overnight, dissected under Voltalef 10S oil and imaged at 40× magnification in a confocal microscope. Cytoplasmic flows were recorded with a 405 nm laser, 4 µs pixel dwell time, collecting 13 frames in 25 s intervals for a total movie duration of 5.17 min.

For the test of *oskar* mRNA localisation under wild-type conditions we used an oskMS, MS2-GFP line (*Belaya and St Johnston, 2011*). For cold experiments, flies were kept at room temperature (22°C), yeast fed overnight at 25°C, and subsequently kept at 25°C for 6 hr before dissection and fixation. In situ hybridizations and imaging of living tissue were performed according to standard methods (*Palacios and St Johnston, 2002*). The *bicoid* mRNA probe was labeled with Digoxigenin-UTP (Boehringer Mannheim).

## Data analysis

To compare topologies of in vivo cytoplasmic flows with topologies of numerically computed cytoplasmic flow fields, we used PIV to generate vector fields capturing the instantaneous direction and magnitude of particle movements. PIV was performed using the open source code PIVlab (*Thielicke and Stamhuis, 2013*), and all PIV vector fields obtained from pairs of consecutive movie frames were averaged over the movie duration. For each movie of cytoplasmic streaming, the mean flow speeds were calculated as the average over all PIV vector magnitudes. Flow speeds obtained from PIV were confirmed by automatic particle tracking using open source code by Blair and Dufresne (*Blair and Dufresne, 2013*). Only particles that could be tracked in at least three consecutive frames were accepted. Note that instantaneous speeds were computed as ratio of particle distance traveled between frames and frame interval. This method avoids underestimation of flow velocities as was pointed out by authors of *Serbus et al. (2005)*, thereby ensuring that our conclusions about the neglibible contribution of flows to transport remain sound.

## Flow fields and oocyte nucleus

### Autocorrelation function

Cytoplasmic flow patterns not only show a high variability between different oocytes, but also change over time in any individual oocyte. To estimate the rate of change of individual flow fields we calculate the unbiased, discrete vector autocorrelation over time for the sequence of PIV flow fields in each movie according to the expression

$$C(k) = \left\langle \frac{1}{N-k} \sum_{i=x,y} \sum_{n=0}^{N-k} (u_i(n+k)\ u_i(n)) \right\rangle, \tag{60}$$

where angular brackets denote the average across all points of the PIV grid, and the peak value of $C(k)$ is normalized to one. Autocorrelation functions from $N = 48$ movies exhibit a large spread across generally low correlation values, and an exponential fit gives a decay time constant of 4.4 min.

## Nucleus to oocyte volume ratio

In 2D confocal images, the oocyte nucleus occasionally appears to cover a significant fraction of oocytes, particularly in young oocytes up to stage 8. To quantify the size of the nucleus compared to the size of the oocyte, we therefore estimate the fraction of the oocyte volume that is covered by the nucleus. We first extract the entire boundary of the oocyte using a custom written macro in Fiji (*Schindelin et al., 2012*). Averaging the boundary shape from below and above the AP axis results in a symmetrized parametric curve denoted by $(x(t), y(t))$ (*Figure 2—figure supplement 1C*, blue curve). After subdividing the boundary into anterior $(x_A(t), y_A(t))$ and posterior $(x_P(t), y_P(t))$ curve, the volume of the respective solids of revolutions can be computed as

$$V_i = \pi \int dt (y_i(t))^2 \frac{dx_i}{dt} \ . \tag{61}$$

While the anterior surface is typically curved inwards for stage 9 oocytes, the anterior surface in young oocytes up to stage 8 can be outwards or inwards curved. Hence, depending on the shape of the anterior the volumes of the anterior $V_A$ and $V_P$ have to be suitably added or subtracted.

The parabolic caps used in the model geometry (*Equation 3*) can be parameterized along the $z$-direction, and in dimensional units their volume is then determined by

$$V_i^\sigma = \pi \int_0^{L\,z_0^i} dz \left( L \sqrt{1 - z/(L\,z_0^i)} \right)^2,$$

$$= \pi L^3 z_0^i / 2 \ . \tag{62}$$

Given that the anterior surface in the standard geometry-1 is curved inwards, the total oocyte volume is given by the difference between between the posterior and the anterior cap volumes $V_P^\sigma - V_A^\sigma = \pi L^3 (z_0^P - z_0^A)/2$.

To approximate the volume of the nucleus, we select its shape and fit a circle to the selection in Fiji. Using the circle radius, we calculate the volume of the nucleus as $4/3\pi r^3$. We find that the nucleus generally covers less than 2.5% of the oocyte volume (*Figure 2—figure supplement 1*), thereby showing that the nucleus volume is negligible even in young stage 9 oocytes, and that 2D confocal images convey a poor impression of 3D volumes. Given the small volume of the nucleus and its location at the anterior of wild type oocytes, the nucleus is unlikely to influence *oskar* mRNP transport towards the posterior, and we therefore neglect it in simulations of cargo transport.

## Impact on flow field

The nucleus may be expected to disturb the cytoplasmic flow field by means of adding an additional no-slip boundary condition inside the volume. We tested this effect by including a spherical excluded volume at the oocyte anterior surface in the solution of the 3D Stokes equations (*Figure 2—figure supplement 1A,F,H*). The nucleus leads to a clear local disturbance of the flow field in a cross section that includes the nucleus compared to the flow field without (*Figure 2—figure supplement 1E,F*). However, taking a perpendicular cross section through the flow fields in which the nucleus is not visible shows that the nucleus rarely changes the flow field (*Figure 2—figure supplement 1G,H*). This low sensitivity with respect to excluded internal volumes is due to the fact that flows are driven by volume rather than by boundary forces. Note that the flow fields shown in *Figure 2—figure supplement 1* still likely overestimate impact of the nucleus because it generally is not spherical and often seen to squeeze deeply into the anterior corners, therefore reaching far less into the oocyte volume than the idealized sphere used here.

## Acknowledgements

This work was supported in part by the Boehringer Ingelheim Fonds and EPSRC (PKT), core support from the Wellcome Trust [092096] and Cancer Research UK [A14492], the MIT Solomon Buchsbaum Award (JD), a Wellcome Trust Principal Research Fellowship [080007] (D St J, HD), the Leverhulme Trust, and the European Research Council Advanced Investigator Grant [247333] (REG).

# Additional information

### Funding

| Funder | Grant reference | Author |
|---|---|---|
| Wellcome Trust | 092096 | Daniel St Johnston |
| European Research Council | 247333 | Philipp Khuc Trong, Raymond E Goldstein |
| Boehringer Ingelheim Fonds | | Philipp Khuc Trong |
| Leverhulme Trust | F/09 618/G | Raymond E Goldstein |
| Cancer Research UK | A14492 | Daniel St Johnston |
| Wellcome Trust | PRF 080007 | Daniel St Johnston, Hélène Doerflinger |
| Engineering and Physical Sciences Research Council | | Philipp Khuc Trong |

The funders had no role in study design, data collection and interpretation, or the decision to submit the work for publication.

### Author contributions

PKT, Conception and design, Acquisition of data, Analysis and interpretation of data, Drafting or revising the article; HD, JD, Acquisition of data, Analysis and interpretation of data, Drafting or revising the article; DStJ, Analysis and interpretation of data, Drafting or revising the article; REG, Conception and design, Analysis and interpretation of data, Drafting or revising the article

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
