## [Decision Letter]

Thank you for sending your work entitled “3D Modelling of microtubules, flows and mRNA transport in *Drosophila* reveals transitions in cytoskeletal organization” for consideration at *eLife*. Your article has been favorably evaluated by Diethard Tautz (Senior editor) and 3 reviewers, one of whom is a member of our Board of Reviewing Editors.

The Reviewing editor and the other reviewers discussed their comments before we reached this decision, and the Reviewing editor has assembled the following comments to help you prepare a revised submission.

This manuscript presents a stochastic theoretical model of microtubule nucleation and growth to study the spatial organization of the 3D cortex of the *Drosophila* oocyte. The key finding of the work is that a simple gradient of microtubule nucleation along the lateral surface naturally leads to the formation of a microtubule network that looks like the one seen in the oocyte and that can guide oscar and unprocessed bicoid mRNA to the specific points in the cell where they are found. The authors also show that cytoplasmic streaming occurs but plays no role in positioning.

This work is interesting and novel. However the referees were not fully convinced that the work is of sufficient general interest for publication in *eLife*. The impression that the work might be mainly of interest for specialists could stem from the presentation of the work which is difficult to read. In particular, it is difficult to understand what exactly are the key findings of the paper. Despite these problems the referees have agreed to give the authors the opportunity to revise their manuscript as the work is very interesting and contributes novel ideas. It provides a systematic approach to determine which transport mechanisms are at work or may be at work. So far this has been quite unclear and there was much speculation. However the authors have to demonstrate in a revised manuscript that their work is of general interest before this submission could be suitable for *eLife*. The authors should revise their manuscript taking into account the points raised below.

Specific criticism:

– in general, this paper is quite hard to digest. A lot of information is provided but the reader is not really well assisted to make sense of this information. It is difficult to extract what it is exactly the authors find out with their work. The logic of the arguments is overall not very clearly laid out, even though the models are well explained. Some of the issues raised below might even be addressed somewhere in the manuscript but I did not find these points clarified.

– The authors show that when averaging over many realizations of the MT network, a clear pattern of direction bias emerges inside the cell (Figure 1). However it remains unclear whether a single realization that would correspond to a real oocyte is good enough. Would the bias in single realization be good enough to position the mRNA properly? I understand that on average the proposed mechanism would work. But this is not the same as saying it works essentially every time. This is a real worry about the paper that the authors do not address.

– The authors calculate cytoplasmic flow patterns using a force distribution obtained from the MT models. In general, flows might influence the cytoskeleton and change its structure. Are there good reasons to neglect such effects? Could such effects be important?

Where I would like to have seen a little more explanation is around [Disp-formula equ1], when the authors introduce the force field f. They vaguely write that it results from motor-driven transport. The authors do not specify what is transported and why the net flow only results from motors of one kind (dyneins and kinesins might segregate and generate flows of different strengths in different compartments of the oocyte). I would like to invite the authors to discuss in more detail the assumptions underlying their choice of f.

– The discussion of “normal” and “untreated” bicoid mRNA is unclear. It is interesting that the model can account for the behavior of injected “untreated” bicoid mRNA. Also the failure of the model to explain the behavior of normal mRNA is very interesting but it also raises many interesting questions. Unfortunately the discussion of these facts is quite unclear and only paragraph 16 of the Result section is this important distinction (normal versus untreated) introduced. In the figure caption this distinction is not mentioned. I think it would help preparing this issue in the Introduction and it is important to discuss the implications of this result carefully.

[Editors' note: further revisions were requested prior to acceptance, as described below.]

Thank you for resubmitting your work entitled “Cortical microtubule nucleation can organise the cytoskeleton of Drosophila oocytes to define the anteroposterior axis” for further consideration at *eLife*. Your revised article has been favorably evaluated by Diethard Tautz (Senior editor) and a Reviewing editor.

There are still some comments from the Reviewing Editor that should be addressed to optimize the final manuscript:

The authors have significantly improved the manuscript. Important points are now clarified and the revised manuscript is much better to read. Before the manuscript can be finally accepted, the authors should consider the following points and suggestions for clarification.

An important point is the statement the MT network “turns over rapidly”.

This clarifies the issue about the possibility of the system to average

over many realizations of stochastic microtubule networks. The authors do

not give an estimate for the turnover time. Rather they imply that since

MT's in the presence of colchicine disappear “within minutes” this suggests “rapid” turnover. However, this is not very clear. Since colchicine is a MT-depolymerizing drug the most simple interpretation could be that MT's disappear because of direct colchicine-triggered depolymerization. Therefore the argument for rapid turnover should be more carefully explained and it would be useful to state what estimated values/ranges for the MT turnover time the authors use.

The authors state in the Results that they “cycled through the pairs of fluid flow and motor-velocities...” It is not very clear what this means. I interpret this statement as describing an alternation between generating a new MT meshwork and solving [Disp-formula equ1] for this network and then during a short time interval solving (2) with constant flow velocity u - and repeating this procedure. However this should be explained. What are the time-intervals chosen and is this choice relevant for the results?

In the Results section sixteenth paragraph, the authors discuss results for naive bicoid simulations shown in Figure 3. It remains unclear here what experimental observations are and whether “splitting” suggested in the model can be seen experimentally. Note also that the sentences in this paragraph are written such that it is sometimes difficult to understand whether they refer to experiments or to calculations. I also found it difficult to find information in the text about the exact differences between naive and endogenous bicoid in experiments.

In the Results section the sentence “However the model does not reproduce transport specifically to the anterior surface when injection is further away from the anterior” is not completely clear. Is it correct to say: “...when injection of conditioned bicoid is further away”? That would be clearer.

---

## [Author Response]

*– in general, this paper is quite hard to digest. A lot of information is provided but the reader is not really well assisted to make sense of this information. It is difficult to extract what it is exactly the authors find out with their work. The logic of the arguments is overall not very clearly laid out, even though the models are well explained. Some of the issues raised below might even be addressed somewhere in the manuscript but I did not find these points clarified*.

Thank you for bringing these general difficulties to our attention. We have significantly reduced the amount of specialist background information provided in the introduction. For example, the description of the MT cytoskeleton prior to the relevant stage 9 and details of the interactions between follicle cells and the oocyte have been removed. Furthermore, each section of the results now ends with a brief summary of the conclusions reached up to this point to clarify and stress the overall logic.

*– The authors show that when averaging over many realizations of the MT network, a clear pattern of direction bias emerges inside the cell (*Figure 1*). However it remains unclear whether a single realization that would correspond to a real oocyte is good enough. Would the bias in single realization be good enough to position the mRNA properly? I understand that on average the proposed mechanism would work. But this is not the same as saying it works essentially every time. This is a real worry about the paper that the authors do not address*.

The microtubules in the oocyte are highly dynamic and a single realization of the MT cytoskeleton would be unrepresentative of the many alternative MT configurations that *oskar* mRNA samples during the 6-9 hour period of its localization. When oocytes are treated with colcemid or colchicine, MTs disappear in minutes (Theurkauf et al., Development 1992; V. Trovisco, personal communication). These drugs block new MT formation by sequestering free tubulin dimers and have no effect on stable MTs. The disappearance of MTs therefore demonstrates that oocyte MTs are dynamic and constantly turn over on a time scale of minutes. Thus, the MT cytoskeleton likely samples many tens to more than a hundred configurations over the period of mRNA localization (6-9 hours). Similarly, cytoplasmic flows are constantly changing over time. Therefore, changing cytoskeletal and flow transport fields in our simulations represents an appropriate approximation of the *in vivo* system.

We have performed simulations of cargo transport with single realizations of the computed MT cytoskeleton and the corresponding flow field. These show a distribution of *oskar* mRNA that changes more slowly than when many realizations are used. While for some pairs of transport fields, *oskar* mRNA appears stuck in the interior of the oocyte, it frequently still moves towards and accumulates at the posterior. The localization is often not centered at the tip of the posterior. This may suggest that MT turnover could refine mRNA localization by averaging out random fluctuations in MT orientations due to low filament number at the posterior. Naïve *bicoid* localizes as before in these simulations. Thus, the bias in a single realization is sufficient to position naive *bicoid* mRNA, and often also correctly positions *oskar* mRNA.

To make this point clear to readers, we have added the following explanation to the text:

*“*However, cargo localisation in the oocyte does not involve only a single cytoskeletal realisation. MTs disappear within minutes in the presence of the MT-depolymerising drug colchicine that blocks MT growth by sequestering free tubulin dimers, indicating that the whole network turns over rapidly (Theurkauf1992, Zhao2012) (V. Trovisco, personal communication). Thus, the oocyte samples many tens to a hundred of independent MT organisations over the 6-9 hours of stage 9.*”*

– The authors calculate cytoplasmic flow patterns using a force distribution obtained from the MT models. In general, flows might influence the cytoskeleton and change its structure. Are there good reasons to neglect such effects? Could such effects be important?

In general, flows might influence cytoskeletal organization by physically reorienting MTs. However, kinesin heavy chain null mutants abolish cytoplasmic flows completely (Palacios and St Johnston, Development 2002) without noticeably changing the MT organisation at stage 9 (Brendza et al., Science 2000). This indicates that the flows are too slow and weak to the affect the MT organization substantially, at least in part because the presence of a cytoplasmic actin mesh restricts the movement of MTs (Dahlgaard et al., Dev Cell 2007). These results suggest that any influence of the flow on the cytoskeleton is likely to be minor and can hence be neglected for our modeling purposes.

We note that flows substantially change MT organization later in oogenesis when the actin mesh disappears or in *capu* and *spire* mutants where the actin mesh does not form at all. In these situations, the MTs visibly reorganize and flows become stronger and more coordinated.

*Where I would like to have seen a little more explanation is around*
[Disp-formula equ1]*, when the authors introduce the force field f. They vaguely write that it results from motor-driven transport. The authors do not specify what is transported and why the net flow only results from motors of one kind (dyneins and kinesins might segregate and generate flows of different strengths in different compartments of the oocyte). I would like to invite the authors to discuss in more detail the assumptions underlying their choice of f*.

Experiments with Kinesin heavy chain null mutants (Palacios and St Johnston, Development 2002; [51]) showed that cytoplasmic flows are entirely dependent on Kinesin. Dynein inhibition has even been reported to slightly increase flow speeds, suggesting a Dynein-mediated inhibition of Kinesin (Serbus et al., Development 2005). As flows are only driven by Kinesin activity, we only consider forces in the direction of MT plus-ends according to vector fields shown in Figure 1 F, G.

The exact force distribution may in general vary spatio-temporally depending on the distribution and motility of the Kinesin-driven cargo that produces the flow. However, the identity of this flow-driving cargo is unknown (it is known, however, that flow does not depend on *oskar* mRNA which being transported in our simulations (Palacios and St Johnston, Development 2002)). To minimize assumptions, we hypothesized that the forces are produced everywhere on the cytoskeleton, thereby making the force distribution directly proportional to the local vector field of the MT cytoskeleton. The scalar factor of proportionality is chosen such as to make the computed flow speeds match the measured flow speed. These aspects have been clarified in the manuscript:

*“In vivo* during stage 9, the oocyte cytoplasm undergoes slow cytoplasmic flows that are abolished in *kinesin heavy chain* mutants. This indicates that flows are driven by Kinesin-dependent transport of an unknown cargo through the viscous cytoplasm (45; 51), *…………..* We make the simplest possible assumption that the forces **f** are proportional to the motor-velocity field, and use experimentally measured flow speeds to calibrate the scalar factor of proportionality.*”*

– The discussion of “normal” and “untreated” bicoid mRNA is unclear. It is interesting that the model can account for the behavior of injected “untreated” bicoid mRNA. Also the failure of the model to explain the behavior of normal mRNA is very interesting but it also raises many interesting questions. Unfortunately the discussion of these facts is quite unclear and only paragraph 16 of the Result section is this important distinction (normal versus untreated) introduced. In the figure caption this distinction is not mentioned. I think it would help preparing this issue in the Introduction and it is important to discuss the implications of this result carefully.

We have simplified and clarified the description of injected *bicoid* (now called naive and conditioned *bicoid*) in the paper. The distinction between both types of injected *mRNA* is now already set up in the introduction, and the legend of Figure 3 clarifies that simulations reproduce naive *bicoid* mRNA localization only. In the Discussion, we describe that the model’s failure to recapitulate conditioned *bicoid* RNA localization can be either due to transport along specific MTs that we do not consider, or due to an unknown mechanism that operates independent of MTs.

*[Editors' note: further revisions were requested prior to acceptance, as described below*.*]*

*An important point is the statement the MT network “turns over rapidly”*.

This clarifies the issue about the possibility of the system to average

*over many realizations of stochastic microtubule networks*. *The authors do*

not give an estimate for the turnover time. Rather they imply that since

*MT's in the presence of colchicine disappear “within minutes” this suggests “rapid” turnover. However, this is not very clear. Since colchicine is a MT-depolymerizing drug the most simple interpretation could be that MT's disappear because of direct colchicine-triggered depolymerization. Therefore the argument for rapid turnover should be more carefully explained and it would be useful to state what estimated values/ranges for the MT turnover time the authors use*.

Colchicine does not depolymerize stable microtubules. In cells with stable microtubules, colchicine has little effect on the microtubule organization. For example, little or no effects are seen on MTs in the follicle cells that surround the oocyte when Colchicine is applied in the same concentration that abolishes MTs in the oocyte within 5-10 minutes. Therefore, most or all of the MTs in the oocyte are dynamic and the MT network in the oocyte hence turns over.

For dynamic MTs, Colchicine prevents growth by binding to and sequestering free tubulin dimers. Tubulin:colchicine complexes also prevent further tubulin addition and increase the catastrophe rate (Mohan et al (2013) PNAS 110:8900-5). Therefore, the time of 5-10 minutes after which microtubules disappear after colchicine addition may appear as lower bound on the turn-over time. On the other hand, Colchicine must diffuse through the follicle cell layer and throughout the large oocyte before it can affect MT dynamics, thereby leaving open the possibility that the MTs may disappear in less than 5-10 minutes if this delivery delay had been absent. In summary, in absence of a systematic experimental analysis that is beyond the scope of this paper, we believe that it is reasonable to assume that the MT cytoskeleton in the oocyte turns over on the time scale of several minutes.

Our arguments are reflected in the revised manuscript as follows:

“MTs disappear after 5-10 minutes in the presence of Colchicine (Theurkauf 1992, Zhao 2012, V. Trovisco, personal communication), a drug that blocks microtubule growth and destabilizes dynamic microtubules, indicating that the whole network turns over within minutes.”

*The authors state in the Results that they “cycled through the pairs of fluid flow and motor-velocities...” It is not very clear what this means. I interpret this statement as describing an alternation between generating a new MT meshwork and solving*
[Disp-formula equ1]
*for this network and then during a short time interval solving (2) with constant flow velocity u - and repeating this procedure. However this should be explained. What are the time-intervals chosen and is this choice relevant for the results?*

As described in the manuscript, we first compute a set of 50 realizations of the cytoskeleton before solving [Disp-formula equ2] for each realization to calculate the corresponding fluid flow field. In this way, we pre-compute 50 pairs of cytoskeleton- and flow-transport fields that can be used subsequently in cargo simulations. During the cargo simulations, pairs of cytoskeleton- and flow-transport fields are chosen from the pre-computed pool one after another in randomized order, thereby cycling through the pool. As stated in Materials and Methods section M4.3, each pair of transport fields is active for a simulated time of 3.6 minutes. We have amended section M4.3 by the statement that the cytoskeleton- and flow-transport fields are pre-computed before the start of the cargo transport simulations.

In the first round of review, we described that oskar mRNA often still localizes to the posterior of the oocyte in simulation with only a single pair of transport fields (i.e. no averaging). In such simulations, the chosen pair of fields remains active for the entire 3 hours of simulated time. This illustrates that the results do not depend critically on the exact choice of the time-interval for the pair of transport fields.

*In the Results section sixteenth paragraph, the authors discuss results for naive bicoid simulations shown in*
Figure 3*. It remains unclear here what experimental observations are and whether “splitting” suggested in the model can be seen experimentally. Note also that the sentences in this paragraph are written such that it is sometimes difficult to understand whether they refer to experiments or to calculations. I also found it difficult to find information in the text about the exact differences between naive and endogenous bicoid in experiments*.

We have clarified the agreement between simulations of bicoid mRNA transport and experimental observations of bicoid injection experiments by expanding the relevant paragraph as follows:

“Simulations show that bicoid mRNA accumulates at both posterior-lateral sides (Figure 3) when initially placed in the posterior half of the oocyte (Figure 3, inset). This is in very good agreement with experimental observations when naive bicoid mRNA is injected into this posterior region [[11] Cell]. When bicoid mRNA is placed initially in the anterior-ventral region (Figure 3, inset), it accumulates at the anterior and lateral cortex (Figure 3), again in concordance with the experimental observations [[11] Cell]. This simulated localisation of bicoid mRNA remains virtually identical in the absence of cytoplasmic streaming. Thus, splitting of a bulk amount of injected bicoid mRNA occurs when the RNA is placed on the border (separatrices) between two diverging subcompartments of the MT cytoskeleton, each one transporting part of the cloud towards the adjacent cortex.”

For protocols for the production of naïve bicoid and its properties we refer the reader to the paper by [11] Cell.

*In the Results section the sentence “However the model does not reproduce transport specifically to the anterior surface when injection is further away from the anterior” is not completely clear. Is it correct to say: “...when injection of conditioned bicoid is further away”? That would be clearer*.

Thank you for this concrete and specific suggestion. We feel, however, that the suggested phrase “injection of conditioned *bicoid*” does not accurately reflect the fact that we do not decide *a priori* about the type of *bicoid* that is injected in the simulations. Instead, in all simulations we merely inject a tracer that moves according to the transport fields and diffusion. From the comparison of simulation results to experimental data, we then infer that this tracer under the conditions used in the simulations behaves like naïve but not like conditioned *bicoid*.